# Improved use of a public good selects for the evolution of undifferentiated multicellularity

**John H Koschwanez[1]\*, Kevin R Foster[2], Andrew W Murray[1]**

[1]FAS Center for Systems Biology and Department of Molecular and Cellular Biology, Harvard University, Cambridge, United States; [2]Department of Zoology, University of Oxford, Oxford, United Kingdom

**Abstract** We do not know how or why multicellularity evolved. We used the budding yeast, *Saccharomyces cerevisiae*, to ask whether nutrients that must be digested extracellularly select for the evolution of undifferentiated multicellularity. Because yeast use invertase to hydrolyze sucrose extracellularly and import the resulting monosaccharides, single cells cannot grow at low cell and sucrose concentrations. Three engineered strategies overcame this problem: forming multicellular clumps, importing sucrose before hydrolysis, and increasing invertase expression. We evolved populations in low sucrose to ask which strategy they would adopt. Of 12 successful clones, 11 formed multicellular clumps through incomplete cell separation, 10 increased invertase expression, none imported sucrose, and 11 increased hexose transporter expression, a strategy we had not engineered. Identifying causal mutations revealed genes and pathways, which frequently contributed to the evolved phenotype. Our study shows that combining rational design with experimental evolution can help evaluate hypotheses about evolutionary strategies.

**\*For correspondence:**
jkoschwanez@cgr.harvard.edu

**Competing interests:** The authors declare that no competing interests exist.

**Reviewing editor**: Diethard Tautz, Max Planck Institute for Evolutionary Biology, Germany

## Introduction

Multicellular organisms have evolved from a unicellular ancestor at least 25 times (*Grosberg and Strathmann, 2007*), but we know little about what selected for the simplest form of multicellularity: an undifferentiated clump of cells produced by the repeated division of a single cell. Two driving forces have been proposed, protection from a variety of factors (including predation [*Kessin et al., 1996*], environmental stress [*Smukalla et al., 2008*], and phagocytosis [*Boraas et al., 1998*]) and more efficient nutrient usage (*Dworkin, 1972*; *Pfeiffer and Bonhoeffer, 2003*; *Koschwanez et al., 2011*; *Alegado et al., 2012*).

In earlier work, we showed that sharing public goods favors clumps over isolated cells and proposed that sharing could have selected for simple multicellularity (*Koschwanez et al., 2011*). The budding yeast, *Saccharomyces cerevisiae*, utilizes sucrose by secreting invertase (*Dodyk and Rothstein, 1964*; *Carlson et al., 1981*). Over 95% of this enzyme remains in the cell wall (*Esmon et al., 1987*; *Tammi et al., 1987*), where it hydrolyzes sucrose into glucose and fructose, which are imported into the cell by a variety of hexose transporters (*Meijer et al., 1996*; *Reifenberger et al., 1997*). Lab yeast strains cannot grow from low density in low concentrations of sucrose because of diffusion: each cell captures only a small fraction of the sugars that sucrose hydrolysis releases, and the molecules released by other, distant cells are at very low concentration. As a result, cells cannot capture enough glucose and fructose to grow. Forming multicellular clumps overcomes this failure; cells in a clump can capture glucose and fructose diffusing from their neighbors and grow in concentrations of sucrose where low concentrations of individual cells cannot.

**eLife digest** Life first appeared on Earth more than 3 billion years ago in the form of single-celled microorganisms. The diverse array of complex life forms that we see today evolved from these humble beginnings, but it is not clear what triggered the evolution of multicellular organisms from single cells.

One of the simplest multicellular eukaryotes is the yeast, *Saccharomyces cerevisiae*—a fungus that has been used for centuries in baking and brewing and, more recently, as a model organism in molecular biology. Yeast cells feed on sugar (sucrose), but are unable to absorb it directly from their surroundings. Instead they secrete an enzyme called invertase, which breaks down the sucrose into simpler components that cells can take up with the help of sugar transporters.

However, single yeast cells living in a low-sucrose environment face a problem: most of the simple sugars that they produce diffuse out of reach. To overcome this difficulty, the cells could form multicellular clumps, which would enable each cell to consume the sugars that drift away from its neighbours. Alternatively, the cells could increase their production of invertase, or they could begin to take up sucrose directly.

Using genetic engineering, Koschwanez et al. produced three strains of yeast, each with one of these traits, and confirmed that all three strategies do indeed help fungi to grow in low sucrose. But could any of these traits evolve spontaneously? To test this possibility, Koschwanez et al. introduced wild-type yeast cells into a low-sucrose environment and studied any populations of cells that managed to survive. Of 12 that did, 11 had acquired the ability to form multicellular clumps, while 10 had increased their expression of invertase. Surprisingly, none had evolved the ability to import sucrose. However, 11 of the populations that survived also displayed an adaptation that the researchers had not predicted beforehand: they all expressed higher levels of the sugar transporters that take up sucrose breakdown products.

The work of Koschwanez et al. suggests that the benefits of being able to share invertase and, therefore, simple sugars, may have driven the evolution of multicellularity in ancient organisms. Moreover, their use of rational design (engineered mutations) combined with experimental evolution (allowing colonies to grow under selection pressure and studying the strategies that they adopt) offers a new approach to studying evolution in the lab.

Speculating on evolution based on experiments with engineered strains is problematic. How well is the ease of engineering a strategy correlated with its evolutionary accessibility? Are multiple mutations required? Do these mutations reduce fitness in other environmental conditions? Are other strategies more accessible? Do certain combinations of strategies outcompete single strategies? And finally, how many different strategies does a set of parallel cultures adopt? Experimentally evolving populations, characterizing their phenotypes, and finding the mutations responsible for evolution allows us to address these questions.

We compared engineered and evolved solutions to the problem of growing on low sucrose concentrations. We engineered and tested three strategies that allow yeast cells to grow on low sucrose: forming multicellular clumps, importing sucrose before hydrolyzing it, and increasing invertase expression. We also evolved unengineered, laboratory strains to grow in the same conditions. All but one of the 12 clones selected from 10 independent populations form multicellular clumps as a result of incomplete cell separation, showing that this selection efficiently selects for multicellularity. In addition, 10 of the clones elevated invertase expression and 11 elevated hexose transporter expression, a strategy that we failed to anticipate, but none showed evidence of sucrose import. We combined bulk segregant analysis (*Michelmore et al., 1991*; *Brauer et al., 2006*; *Segrè et al., 2006*; *Birkeland et al., 2010*; *Magwene et al., 2011*) with whole genome sequencing to identify putative causal mutations. We recreated two of the evolved clones, one with five mutations and one with eight mutations, to show that mutations we had identified were indeed causal. Finally, we competed the evolved clones against their ancestor and found that adaptation in sucrose severely reduces fitness in high glucose.

## Results

### Three engineered strategies for growth in low sucrose

We began by asking whether undifferentiated multicellularity was the only strategy that allowed yeast cells to grow on low sucrose. We hypothesized that there were two alternative strategies: increasing invertase expression, and importing sucrose and then hydrolyzing it inside the cell (*Figure 1*). Having previously tested undifferentiated multicellularity (*Koschwanez et al., 2011*), we engineered the other two strategies and tested whether they allow cells to grow from low densities in low sucrose concentrations. Population growth requires both cell growth and cell proliferation. For simplicity, we refer to the combination of these properties as growth.

Increased invertase expression will increase the rate of sucrose hydrolysis at the cell wall. Although cells will still lose most of the fructose and glucose to the environment, the increased hydrolysis rate will increase the monosaccharide concentration at the plasma membrane and lead to a higher rate of sugar import. We increased invertase expression by replacing the promoter of the invertase gene (*SUC2*) with the *GAL1* promoter in a yeast strain that is unable to utilize galactose (see *supplementary file 3* for all strains used in this study; *Ingolia and Murray 2007*). As a result, galactose serves as a gratuitous inducer: it induces Suc2 but cannot itself be metabolized. Single cells were placed in each microwell of a 96-well plate, and incubated with a range of sucrose and galactose concentrations. *Figure 2A* shows that increasing invertase expression allows growth from a single cell in low sucrose concentrations; increasing inducer concentrations leads to better growth.

An alternative strategy is importing sucrose before hydrolyzing it since some invertase molecules are retained in the cytoplasm rather than being exported by protein secretion. Mal11 is a maltose importer that can also import sucrose (*Stambuk et al., 1999*), but is not expressed in our yeast strains

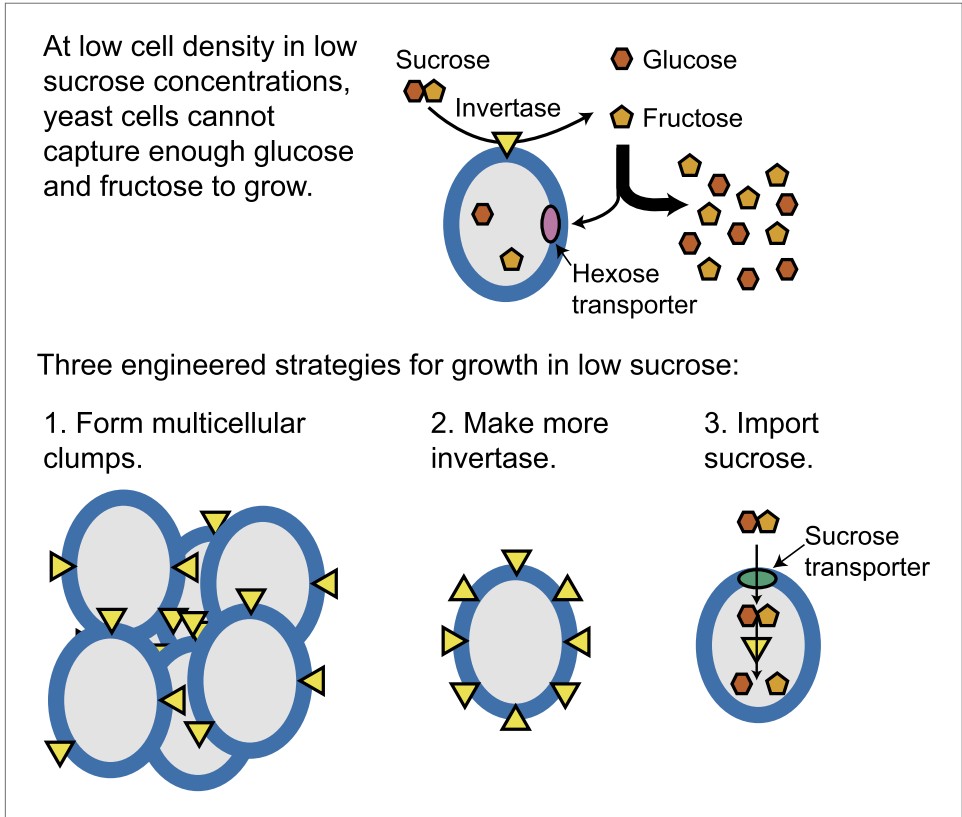

**Figure 1**. Three engineered strategies for growth in low sucrose. Strategy 1, form multicellular clumps, was previously verified (*Koschwanez et al., 2011*). The results of testing strategy 2, make more invertase, and strategy 3, import sucrose, are shown in *Figure 2*. All three strategies outcompete wild-type strains when the sole carbon source is 1 mM sucrose (*Table 1*).

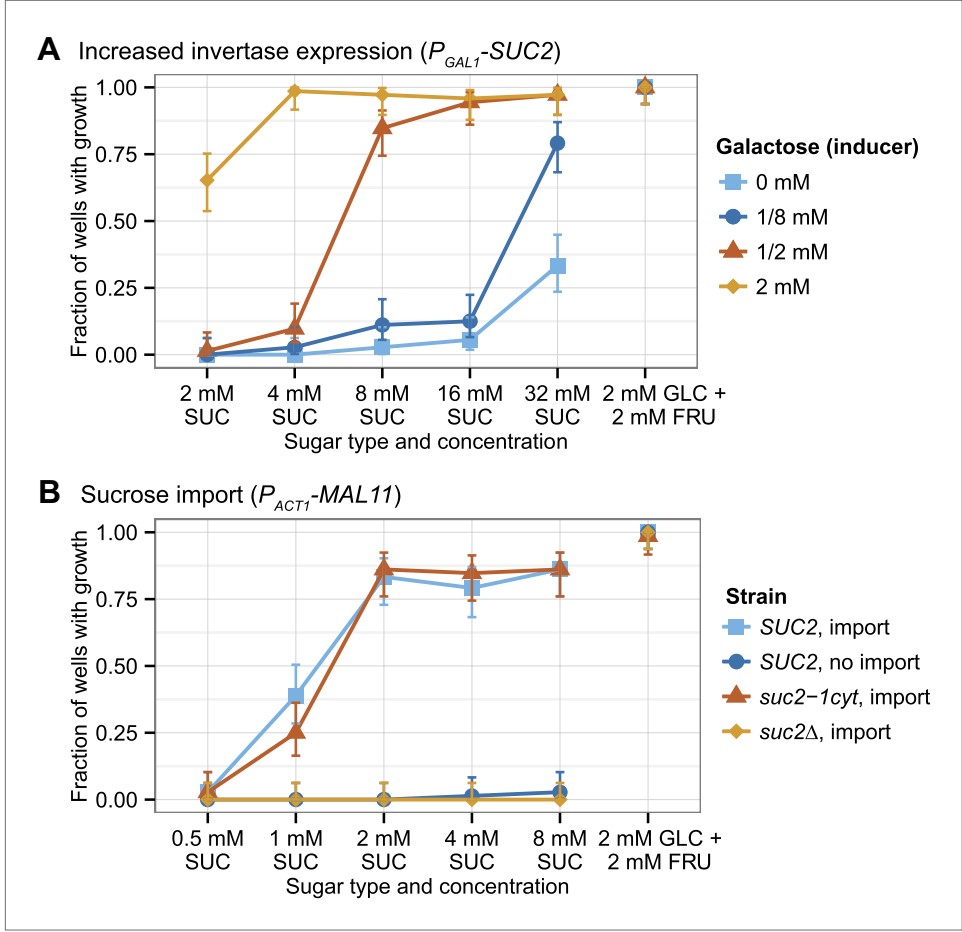

**Figure 2**. Two strategies for growth from low sucrose concentrations. (**A**) Strong expression of secreted invertase allows growth from a single cell at low sucrose concentrations. All *GAL1* promoter induction data is from the same yeast strain yJHK312 in which transcription of *SUC2* is driven by the *GAL1* promoter. Galactokinase (*GAL1*) is deleted from this strain so that galactose acts as an inducer and not as a carbon source, and the Gal regulon has been engineered to produce a graded rather than a bistable response to increased galactose concentrations by overexpressing *GAL3* from the *ACT1* promoter (***Ingolia and Murray, 2007***). (**B**) Sucrose import allows growth from a single cell in low sucrose concentrations. The '*SUC2*, import' strain yJHK372 expresses *SUC2* from the *SUC2* promoter and *MAL11* from the *ACT1* promoter. The '*SUC2*, no import' strain yJHK222 expresses *SUC2* from the *SUC2* promoter. The '*suc2-1cyt*, import' strain yJHK373 expresses cytoplasmic invertase from the *SUC2* promoter and *MAL11* from the *ACT1* promoter. The '*suc2Δ*, import' strain yJHK374 has *SUC2* deleted and expresses *MAL11* from the *ACT1* promoter. For both (**A**) and (**B**), single cells were inoculated by fluorescence activated cell sorting (FACS) into 150 μl wells at the given sugar and galactose concentration and grown without shaking for 85 hr at 30°C and the results shown are totals of three experiments; each experiment used one plate per sugar concentration, and each plate used 24 wells per strain or galactose concentration. In both figures, 2 mM glucose + 2 mM fructose is used as a positive control, and error bars refer to 95% binomial confidence interval using the adjusted Wald method. FRU is fructose, GLC is glucose, and SUC is sucrose.

(***Brown et al., 2010***). We expressed Mal11 from a strong, constitutive promoter (*P_{ACT1}*) in three different strains: a standard lab strain (*SUC2*), a strain that does not secrete invertase but produces cytoplasmic invertase (*suc2-1cyt*) (***Koschwanez et al., 2011***), and a strain that lacks invertase (*suc2Δ*). *Figure 2B* shows that both the *SUC2* and *suc2-1cyt* strains that make the maltose importer can grow from single cells on as little as 1 mM sucrose. The nearly identical growth of the *SUC2* and *suc2-1cyt* strains shows that the importer makes extracellular sucrose digestion dispensable, and the failure of the *suc2Δ* strains to grow shows that sucrose utilization after import still requires invertase. Thus three different strategies, clumping, increased invertase expression, and sucrose import, each allows yeast to grow from low cell density at low sucrose concentrations.

We competed each of the three strategies against an unmodified strain to ask whether they could invade an ancestral population. Derivatives of the two competing strains, each expressing a different fluorescent protein, were mixed together and passaged on 1 mM sucrose. In each passage, the cells were grown together for almost eight generations and then diluted 200-fold into fresh medium. The competition was assessed by the number of passages required to eliminate the less fit strain, or the ratio between the two strains at the end of the sixth passage. The three engineered strains all outcompeted the ancestral wild-type lab strain in 1 mM sucrose (*Table 1*), although the strain engineered to express increased invertase (EngHiInvertase) was a much worse competitor than the other two (EngClumpy, EngSucImport).

## Experimental evolution produces multicellularity

Having shown that engineering can produce three different strategies for growth on low sucrose, we asked, 'What would evolution do?' We experimentally evolved multiple, parallel cultures to grow well

**Table 1.** Fitness of engineered strains, evolved clones, recreated strains, and reverted strains

| Strain 1 | Strain 2 | 1 mM sucrose | 1 mM glucose + 1 mM fructose | 80 mM glucose |
|---|---|---|---|---|
| EngClumpy | Wild-type lab | +++ | 0 | 0 |
| EngHiInvertase | Wild-type lab | + | 0 | 0 |
| EngSucImport | Wild-type lab | +++ | 0 | 0 |
| EvoClone1 | Ancestor | ++++ | ――― | ―――― |
| EvoClone2 | Ancestor | ++++ | 0 | ―― |
| EvoClone3 | Ancestor | ++++ | ――― | ―――― |
| EvoClone4 | Ancestor | ++++ | + | ――― |
| EvoClone5 | Ancestor | ++++ | + | ――― |
| EvoClone6 | Ancestor | ++++ | + | ―― |
| EvoClone7A | Ancestor | ++++ | ―― | ――― |
| EvoClone7B | Ancestor | ++++ | + | ― |
| EvoClone7C | Ancestor | ++++ | ――― | ―――― |
| EvoClone8 | Ancestor | ++++ | ― | ―――― |
| EvoClone9 | Ancestor | ++++ | ― | ――― |
| EvoClone10 | Ancestor | ++++ | ― | ―――― |
| Recreated2 | Ancestor | ++++ | ― | ― |
| Recreated9 | Ancestor | ++++ | 0 | ― |
| EvoClone2 | Recreated2 | + | + | 0 |
| EvoClone9 | Recreated9 | + | ― | ――― |
| EvoClone2 | Reverted2 | ++++ | + | ― |
| EvoClone9 | Reverted9 | ++++ | + | + |
| EvoClone2 | EngClumpy | ++ | 0 | ―― |
| *ace2Δ* | Ancestor | +++ | 0 | 0 |
| *gin4-W19\* irc8-G57V mck1-G227Vfs249* | Ancestor | +++ | ― | ― |

++++Strain 1 eliminates strain 2 in 1–2 growth cycles.
+++ Strain 1 eliminates strain 2 in 3–4 growth cycles.
++ Strain 1 eliminates strain 2 in 5–6 growth cycles.
+ Strain 1 > 75% of population after 6 growth cycles.
0 Neither strain is >75% of population after 6 growth cycles.
― Strain 2 > 75% of population after 6 growth cycles.
―― Strain 2 eliminates strain 1 in 5–6 growth cycles.
――― Strain 2 eliminates strain 1 in 3–4 growth cycles.
―――― Strain 2 eliminates strain 1 in 1–2 growth cycles.
Growth cycle numbers are averaged over three independent experiments.

on low sucrose. Ten independent populations of budding yeast were serially diluted in minimal, 1 mM sucrose-containing medium. The starting strain for each population was a haploid, non-clumpy, prototrophic strain (*Figure 3A*) that constitutively expressed YFP and carried a DNA polymerase mutation (*POL3-L523D*) that elevated its mutation rate roughly 100-fold (*Jin et al., 2005*). We used a mutator strain to increase the speed of adaptation. At each growth cycle, $5 \times 10^5$ cells were inoculated into 50 ml of medium (*Figure 3B*). Over 25–35 serial dilutions, the time it took the culture to become cloudy fell from 2 weeks to 3 days, and all 10 populations (named EvoPopulation1–10) displayed a clumpy phenotype (*Figure 3C*). There was no loss of constitutive YFP expression.

We asked which strategies individual populations had adopted. Eight clones were selected from each population and their morphology and growth on sucrose were examined. For nine of the ten populations (all but EvoPopulation7), all the clones were morphologically identical; one clone was selected from each of these populations and used for further studies. EvoPopulation7 produced three different phenotypes: one had medium-sized clumps (EvoClone7A), one was not clumpy (EvoClone7B), and one had large clumps (EvoClone7C). Images of the clones are shown in *Figure 3—figure supplement 1*, and size distributions of each clone are shown in *Figure 3—figure supplement 2*.

We tested the ability of the evolved clones to compete with their ancestor. Each of the evolved clones strongly outcompeted their ancestor, eliminating it from cultures within two passages on 1 mM sucrose (*Table 1*). We conclude that efficient use of public goods, liberated through extracellular hydrolysis, selects for the evolution of undifferentiated multicellularity.

## Clumps form by failure in cell separation

We asked how the evolved clones formed clumps. Wild yeast isolates can form multicellular clumps in two ways: flocculation (*Guo et al., 2000*), in which separate cells stick to each other via cell wall-bound adhesins, or failure to separate daughters from their mothers because the cell wall that joins

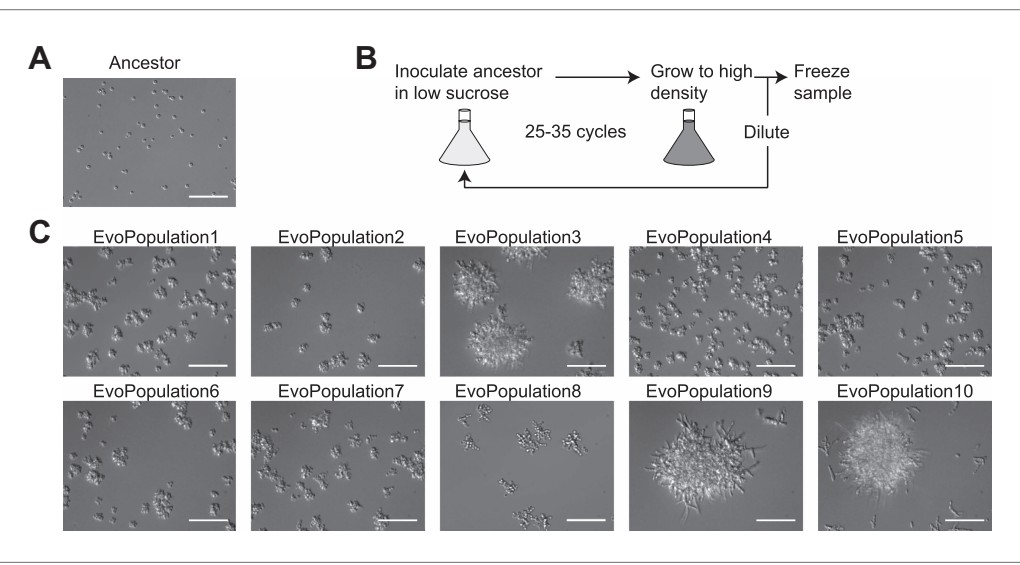

**Figure 3**. Evolved populations show a clumpy phenotype. (**A**) An ancestor derivative (yJHK111) after growth in 1 mM sucrose. (**B**) Schematic of experimental evolution. Cells were inoculated in 1 mM sucrose media, grown to high density, and then $10^5$ cells were reinoculated into fresh media for a total of 25–35 cycles. A sample was frozen down at each serial dilution. (**C**) Samples taken from the last time point of the evolved populations. Representative DIC images were taken with a 40× objective in a glass-bottomed, 96-well plate. All scale bars are 50 µm.

The following figure supplements are available for figure 3:

**Figure supplement 1**. Eleven of twelve clones show a clumpy phenotype.

**Figure supplement 2**. Size distribution of evolved clones.

**Figure supplement 3**. EvoClone9 morphology changes in different media.

them is not digested after cytokinesis (*Yvert et al., 2003*). To find which method the evolved clones used, we co-cultured two different colored versions of each evolved clone, using constitutive expression of different fluorescent proteins to mark the two versions. If the clumps formed by flocculation, many clumps would contain cells of both colors; if cells fail to separate, each clump would contain only one color since it would arise by the repeated division of a single cell.

All the multicellular clones are the result of incomplete separation; each clump contains cells of only one or the other color. Representative images from two evolved clones are shown in *Figure 4* (images from the remaining clones are shown in *Figure 4—figure supplement 1*). We checked our method by using two control strains: a flocculating strain that expresses a high level of Flo1, a known adhesin, (*Smukalla et al., 2008*) and a lab strain that contains a wild isolate allele of *AMN1* (*AMN1-RM11*), which prevents cell separation after many of the cell divisions (*Yvert et al., 2003*). *Figure 4* shows that the flocculating clumps from the strain expressing Flo1 contain both colors, whereas clumps from the *AMN1-RM11* strain contain only one color.

## RNA sequencing shows that most clones elevate invertase expression and hexose transporter expression

We used RNA sequencing to examine the pattern of gene expression in the evolved clones. We isolated and sequenced RNA from the 12 evolved clones and two independent ancestor derivatives, all grown in 1 mM sucrose to log phase. In 10 of the 12 clones invertase (*SUC2*) expression was elevated between 3- and 21-fold above their ancestor (*Table 2*); EvoClone7C and EvoClone8 were the

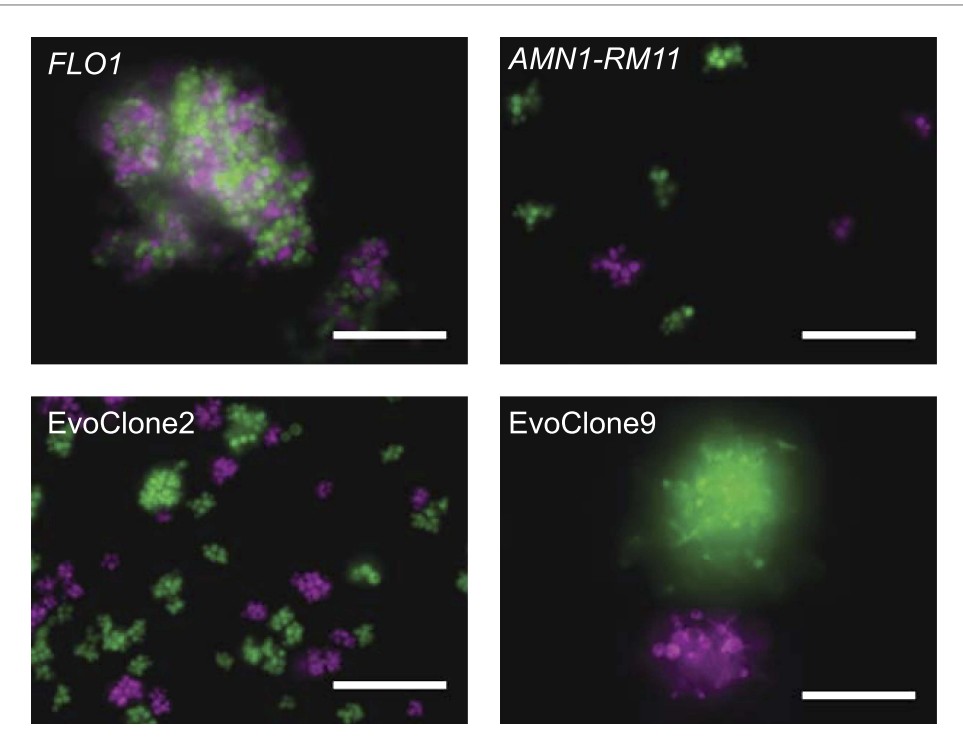

**Figure 4**. Clumpiness is due to failure to separate and not flocculation. Each image shows two genetically identical strains that are labeled with different fluorescent proteins, shown as magenta and green in the image. The strains were grown together from low density in 1 mM sucrose. (**A**) Lab strain with constitutively expressed *FLO1*. Flocculation is evident from the mix of colors in a single clump. (**B**) Lab strain with the RM11 allele of *AMN1*. (**C**) EvoClone2. (**D**) EvoClone9. The clumps in the *AMN1-RM11* strain and the evolved clones are uniform in color, showing that clumpiness is due to failure to separate after cell division. Representative fluorescent images were taken with a 20× objective in a glass-bottomed, 96-well plate. All scale bars are 50 μm.
The following figure supplements are available for figure 4:

**Figure supplement 1**. Clumpiness is due to failure to separate and not flocculation.

**Table 2.** Significant changes in invertase and hexose transporter expression

| Strain name | Significant increases in invertase (*SUC2*) expression | Significant increases in hexose transporter expression |
|---|---|---|
| EvoClone1 | 5X | *HXT1* 42X |
| | | *HXT2* 5X |
| | | *HXT3* 108X |
| | | *HXT4* 91X |
| EvoClone2 | 7X | *HXT4* 9X |
| EvoClone3 | 7X | *HXT4* 4X |
| EvoClone4 | 9X | *HXT1* 4X |
| | | *HXT3* 32X |
| | | *HXT4* 14X |
| | | *HXT9* 5X |
| | | *HXT11* 4X |
| EvoClone5 | 7X | *HXT1* 4X |
| | | *HXT2* 8X |
| | | *HXT3* 6X |
| | | *HXT4* 21X |
| | | *HXT6* 7X |
| | | *HXT7* 6X |
| EvoClone6 | 9X | *HXT2* 9X |
| | | *HXT4* 12X |
| EvoClone7A | 3X | *HXT2* 7X |
| | | *HXT3* 2X |
| | | *HXT4* 92X |
| EvoClone7B | 3X | *HXT2* 10X |
| | | *HXT3* 22X |
| | | *HXT4* 103X |
| EvoClone7C | Not significant | *HXT1* 3X |
| | | *HXT2* 4X |
| | | *HXT3* 10X |
| | | *HXT4* 52X |
| | | *HXT6* 4X |
| EvoClone8 | Not significant | *HXT2* 6X |
| | | *HXT3* 5X |
| | | *HXT4* 13X |
| EvoClone9 | 4X | *HXT2* 5X |
| EvoClone10 | 21X | None |

See ***supplementary file 1C*** for list of all genes that were significantly changed and their change in each evolved clone.

only clones with no significant change in *SUC2* expression.

RNA sequencing revealed another change that likely led to better growth in 1 mM sucrose: hexose transporter expression was elevated in 11 of 12 evolved clones. Yeast encodes at least 16 different hexose transporters, most encoded by members of the *HXT* gene family. The expression of *HXT4*, which encodes a high-affinity glucose transporter, was elevated in 10 of these evolved clones (***Table 2***); EvoClone10 was the only strain without increased expression of any hexose transporter.

The effect of evolution on genes with known roles in cell separation is shown in ***Table 3***. Four genes involved in cell separation, *AMN1*, *CTS1*, *DSE2*, and *SCW11*, had significantly reduced levels in all 11 clumpy evolved clones, and three other genes implicated in cell separation showed reduced expression in 10 (*DSE1, SUN4*), or 9 (*DSE4*) of the clumpy clones. These seven genes are not decreased in expression in the non-clumpy evolved clone (EvoClone7B) or the control strain. This supports the argument that the clumps form by failure to separate. ***Supplementary file 1A*** lists genes whose expression was elevated or reduced in nine or more evolved clones (and unchanged in the control strain). We suspect many of these genes may show differences in expression because the evolved clones grow much more rapidly than their ancestors in low sucrose medium. ***Supplementary file 1B*** lists the genes whose expression was increased or decreased at least tenfold in each of the ten evolved clones we examined. All of the approximately 1500 genes with significant expression level changes are listed in ***supplementary file 1C*** along with their level of change in each evolved clone.

## All evolved clones depend on secreted invertase

Forming multicellular clumps and increasing hexose transporter expression both suggest that evolved clones are still hydrolyzing sucrose extracellularly. But since we had shown that engineering sucrose import allowed growth on low sucrose, we needed to rule out the possibility that some of the evolved clones were using this strategy. If an evolved clone depended on sucrose import, keeping it from secreting invertase would have little effect on its growth in sucrose. To test for this possibility, we removed the signal sequence (***Kaiser and Botstein, 1986***; ***Perlman et al., 1986***) that directs Suc2's secretion from each of the evolved clones (now named EvoCloneX-*suc2-1cyt*) and competed these derivatives against the

**Table 3.** Cell separation genes whose expression fell significantly

| Gene | Reduced in multicellular clones | Reduced in single cell clone | Function |
|---|---|---|---|
| AMN1 | 11/11 | 0/1 | Cell separation protein |
| CTS1 | 11/11 | 0/1 | Cell separation, chitinase |
| DSE2 | 11/11 | 0/1 | Cell separation, possible glucanase |
| SCW11 | 11/11 | 0/1 | Cell separation, possible glucanase |
| DSE1 | 10/11 | 0/1 | Cell separation, protein of unknown function |
| SUN4 | 10/11 | 0/1 | Cell separation, possible glucanase |
| DSE4 | 9/11 | 0/1 | Cell separation, possible glucanase |

corresponding, unmodified, evolved clone. In each case, the evolved clone quickly outcompeted the version that did not secrete invertase (*Table 4*), demonstrating that growth of all the evolved clones depends on secreted invertase. To verify that this growth defect was not due to reduction in invertase expression, we measured, using reverse transcription followed by quantitative PCR (RT-qPCR), the *SUC2* expression of the two evolved clones (EvoClone2 and EvoClone9) and their *suc2-1cyt* counterparts grown in 1 mM glucose. In both cases, expression of *SUC2* was slightly greater in the *suc2-1cyt* strain but statistically insignificant over three independent trials.

## Evolved clones have poor fitness on other carbon sources

Did evolving to grow faster on low sucrose affect the ability of the evolved clones to grow on other carbon sources? Other experiments have resulted in antagonistic pleiotropy, where improved fitness

**Table 4.** Fitness of evolved clones after removal of **SUC2** signal sequence

| Strain 1 | Strain 2 | 1 mM sucrose | 1 mM glucose + 1 mM fructose |
|---|---|---|---|
| EvoClone1 | EvoClone1-*suc2-1cyt* | ++++ | 0 |
| EvoClone2 | EvoClone2-*suc2-1cyt* | +++ | 0 |
| EvoClone3 | EvoClone3-*suc2-1cyt* | ++++ | 0 |
| EvoClone4 | EvoClone4-*suc2-1cyt* | +++ | 0 |
| EvoClone5 | EvoClone5-*suc2-1cyt* | ++++ | 0 |
| EvoClone6 | EvoClone6-*suc2-1cyt* | ++++ | + |
| EvoClone7A | EvoClone7A-*suc2-1cyt* | +++ | 0 |
| EvoClone7B | EvoClone7B-*suc2-1cyt* | +++ | 0 |
| EvoClone7C | EvoClone7C-*suc2-1cyt* | ++++ | + |
| EvoClone8 | EvoClone8-*suc2-1cyt* | ++++ | + |
| EvoClone9 | EvoClone9-*suc2-1cyt* | ++++ | 0 |
| EvoClone10 | EvoClone10-*suc2-1cyt* | ++++ | + |
| wt | wt-*suc2-1cyt* | +++ | 0 |
| wt-*suc2-1cyt*-importer | wt-*suc2-1cyt* | ++++ | 0 |

See *Table 1* for definition of fitness measurements. We used two control competitions: in the first, a standard lab strain outcompeted a lab strain with a missing *SUC2* secretion signal sequence (*suc2-1cyt*); in the second, a *suc2-1cyt* strain with *MAL11* expressed from the *ACT1* promoter outcompeted the *suc2-1cyt* strain that did not express a sucrose importer.

in one environment often corresponds to reduced fitness in another (*Lenski, 1988*; *Wenger et al., 2011*). We therefore determined the fitness of the evolved clones in three environments: low (1 mM) sucrose, low monosaccharide (1 mM glucose plus 1 mM fructose, the hydrolysis products of 1 mM sucrose), and high (80 mM) glucose.

Each evolved clone quickly outcompeted the ancestor in low sucrose and lost to its ancestor in high glucose (*Table 1*). Four of the evolved clones also lost quickly on low monosaccharide; the remaining clones had approximately the same fitness as the ancestor, suggesting that most of the mutations acquired in low sucrose were selectively neutral in the equivalent monosaccharide concentration. The size distributions of each evolved clone was similar in all three conditions (*Figure 3—figure supplement 2*) except for EvoClone9, which formed much larger clumps in sucrose than it did in the other two media (*Figure 3—figure supplements 2* and *3*).

## Putative causal mutations found through bulk segregant analysis

We began the genetic characterization of the evolved clones by looking for the causal mutations. Because we used a strain that produced roughly one mutation per cell cycle, we predicted that neutral or slightly deleterious mutations would substantially outnumber the mutations that made cells grow faster in low sucrose concentrations. The neutral and nearly neutral mutations accumulate because they occur in lineages that were lucky enough to have strongly beneficial mutations; in the absence of sex, these mutations thus hitchhike during selection.

To identify the causal mutations, we used bulk segregant analysis, which uses sexual reproduction to separate causal from hitchhiking mutations. The evolved strain was crossed to its ancestor, put through meiosis, and a large population of haploid spores was selected for the evolved phenotype. Mutations that confer a strong advantage on low sucrose will be present in almost all the selected spores, whereas those that do not will be present in roughly half the spores. We monitored the allele frequency in the selected pool by preparing DNA and sequencing it to roughly 100-fold coverage (*Figure 5*). We found a total of 80 putative causal mutations in the twelve evolved clones out of 1521 mutations, confirming that most mutations are non-causal (*Table 5*). To track the spread of the putative causal mutations through the evolved populations, we used Sanger sequencing to measure the allele frequency (*Gresham et al., 2008*) over time. *Figure 6* shows the spread of putative causal mutations through two of the populations, EvoClone2 and EvoClone9; *Figure 6—figure supplements 1* and *2* show the remaining populations.

Over all 12 clones, the 80 putative causal mutations lie in or near 53 genes. Two genes, *ACE2* and *UBR1,* were mutated in at least half the clones, and groups of genes in three pathways, involved in glucose sensing (the *RGT1* group), growth regulation (the *IRA1/IRA2* group), and transcription (the Mediator group) were also mutated in many clones. We suspect that most of the putative causal mutations are loss of function mutations. Many alleles of the two most frequently mutated genes, *ACE2* and *UBR1* were nonsense mutations. In addition, amongst the 39 genes that were only mutated once, 12 (31%) of the mutations were nonsense mutations. This suggests that many of the mutations in the remaining 27 genes are loss of function mutations: for two genes that have been extensively studied, *URA3* and *CAN1*, 31% of strong loss of function mutations are nonsense mutations (*Lang and Murray, 2008*). (Because loss-of-function mutations are typically recessive, we use lower case italic nomenclature to indicate the putative causal allele (e.g., *irc8-G57V*), and upper case italics to indicate the wild-type allele (e.g., *IRC8*). See the notes for *Table 5* for a description of the mutation nomenclature. Roman text (e.g., Irc8) indicates the protein). *Supplementary file 2* is a summary of the mutations in each gene pathway.

Ace2 activates the transcription of enzymes that degrade the septum that connects mother to daughter cell (*Colman-Lerner et al., 2001*; *Sbia et al., 2008*). *ACE2* was mutated in eight evolved clones; six of these mutations were nonsense mutations scattered through the open reading frame. All six of these clones had at least tenfold reductions in the expression of three genes known to aid in cell separation (*CTS1* [chitinase], *DSE1*, and *DSE2* [*supplementary file 1B*]) in agreement with the original finding that *ace2Δ* mutants are clumpy (*Dohrmann et al., 1992*).

Ubr1 is an E3 ubiquitin ligase in the N-end rule pathway. Ubr1 targets proteins with certain N-terminal amino acids, which are exposed by proteolytic cleavage or removal of the N-terminal methionine, for degradation (*Bartel et al., 1990*). *UBR1* was mutated in 6 of the 12 clones; four mutations were nonsense mutations, strongly suggesting that the remaining two are also a loss of function mutations. Rad6, the ubiquitin-conjugating enzyme that forms a heterodimer with Ubr1 (*Dohmen et al., 1991*), had a missense mutation in a seventh evolved clone.

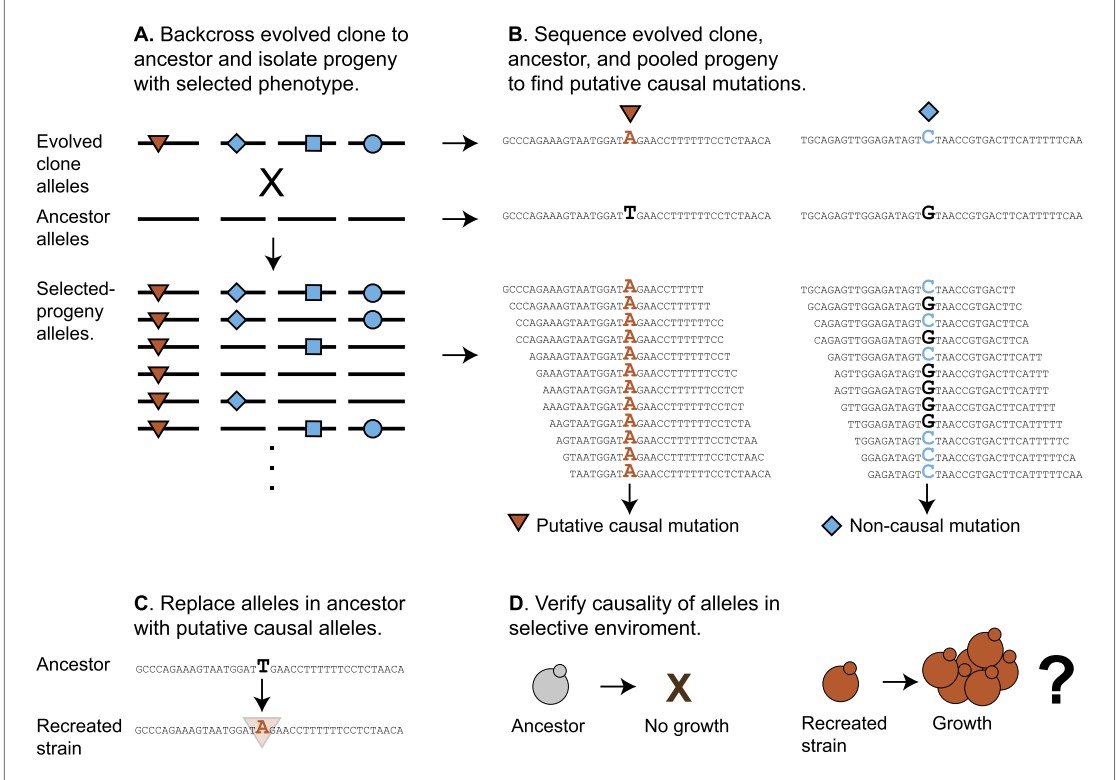

**Figure 5**. Schematic of bulk segregant analysis and evolved clone reconstruction. (**A**) A clone is selected from the population and then backcrossed to a derivative of its ancestor. The resulting diploid is sporulated, allowing the mutant alleles to randomly segregate among the haploid progeny. When the haploid progeny are selected for growth in low sucrose, only those cells with causal alleles (red triangles) remain; non-causal alleles (blue diamond, square, and circle) segregate randomly and are present in about half of the spores. (**B**) The ancestor, evolved clone, and pool of selected progeny are sequenced. Comparing the genome sequences of the ancestor and evolved clone reveals mutations. The allele frequency in the selected spores can then be estimated from the frequency of the reads in the pool of selected progeny. We classified any mutant allele present in >90% of the reads as a putative causal mutation (**Table 5**). (**C**) The wild-type alleles in the ancestor were replaced with the putative causal mutations to recreate the evolved clone (**Figure 7**). (**D**) Growth of the recreated strain was tested in low sucrose (**Table 1**).

The following figure supplements are available for figure 5:

**Figure supplement 1**. Protocol for replacing alleles in yeast.

The Rgt1 pathway controls cells' response to external glucose concentrations. In high glucose, Rgt1, a transcription factor, represses the expression of high affinity hexose transporters such as *HXT4* (***Ozcan and Johnston, 1999***). In low glucose, Snf3, a membrane-spanning glucose sensor, induces degradation of Mth1, a protein that is needed for Rgt1 to extert its repressive effects (***Broach, 2012***). One of the genes in the Rgt1 pathway was mutated in 8 of the 12 clones: *SNF3* (3 missense), *MTH1* (2 nonsense), and *RGT1* (2 nonsense, 1 missense); no clones had two mutations in the pathway. Because Mth1 and Rgt1 work together to repress genes involved in growth on poor carbon sources, we expect that the mutations we isolated are loss of function mutations. *MTH1* loss of function mutations have been selected by others in glucose-limited chemostats (***Kao and Sherlock, 2008***; ***Kvitek and Sherlock, 2011***). On the other hand, because Snf3's activity leads to the induction of glucose-repressed genes, we speculate that the *SNF3* mutations may result in gain of function. One of our mutations, *snf3-A231D*, lies near a known dominant mutation, *SNF3-R229K* (***Ozcan et al., 1996***), that elevates hexose transporter expression.

The Ras-cAMP pathway regulates cell growth in response to nutrients (***Broach, 2012***). Ras1 and Ras2 are small G proteins whose activity responds to nutrient sensing and who stimulate the activity of adenyl cyclase, which produces cAMP. Ira1 and Ira2 both encode GTPase activating proteins that inactivate Ras by stimulating its intrinsic GTPase activity (***Tanaka et al., 1989***, *1990*). *IRA1* or *IRA2* were

**Table 5.** Putative causal mutations in each evolved clone

| Strain name | Nominal generations | Number of mutations | Non-synonymous and promoter mutations segregating at evolved allele frequency >90% | Nucleotide change | Amino acid change | Mutant allele reads/total reads |
|---|---|---|---|---|---|---|
| EvoClone1 | 307 | 71 | ACE2 | 703 G→T | E235* | 77/79 |
| | | | IRA1 | 8987C→G | T2996S | 99/102 |
| | | | PHO87 | 196 G→C | A66P | 55/55 |
| | | | RGT1 | 3157C→T | Q1053* | 72/74 |
| | | | SAN1 | 1707C→A | N569K | 76/77 |
| | | | SIN4 | 383 G→A | G128D | 106/106 |
| | | | UBR1 | 1916T→A | L639* | 42/44 |
| EvoClone2 | 273 | 115 | ACE2 | 968T→A | L323* | 110/110 |
| | | | CSE2 | 100_100delT | S35Rfs54 | 102/107 |
| | | | IRA1 | 7657A→T | S2553C | 72/75 |
| | | | MTH1 | 459_459delC | H154Tfs156 | 93/100 |
| | | | UBR1 | 524 G→A | C175Y | 104/112 |
| EvoClone3 | 301 | 252 | IRA2 | 8081C→A | S2694Y | 75/81 |
| | | | IRC8 | 365T→C | L122P | 84/88 |
| | | | NAT1 | 1782 G→A | W594* | 84/87 |
| | | | SYP1 | 1376C→T | T459I | 90/96 |
| EvoClone4 | 229 | 95 | ACE2 | 670 G→T | E224* | 93/93 |
| | | | RGT1 | 2494_2495insT | L832Ffs834 | 135/139 |
| | | | SIN4 | 382 G→A | G128S | 104/105 |
| | | | UBR1 | 1916T→A | L639* | 56/61 |
| EvoClone5 | 237 | 120 | ACE2 | 565C→T | Q189* | 149/152 |
| | | | ARO2 | 371C→G | A124G | 93/101 |
| | | | MCK1 | 38_38delG | G14Dfs22 | 152/158 |
| | | | SNF2 | 71 G→T | R24I | 94/100 |
| | | | SNF3 | 1235T→A | V412E | 148/148 |
| EvoClone6 | 232 | 110 | ACE2 | 507_507delT | N169Kfs177 | 142/146 |
| | | | GCN2 | 892A→G | N298D | 115/117 |
| | | | GPB2 | 235 G→T | E79* | 144/145 |
| | | | MTH1 | 152_152delG | S51Ifs56 | 110/112 |
| | | | NRG1 | 371C→A | S124* | 92/95 |
| | | | RAD6 | 191C→A | P64H | 127/135 |
| EvoClone7A | 242 | 105 | ACE2 | 1901C→A | S634* | 84/87 |
| | | | RAD61 | 225T→A | N75K | 97/97 |
| EvoClone7B | 242 | 94 | GCN3 | 176C→A | S59Y | 79/87 |
| | | | IRA2 | 7049_7049delC | A2350Gfs2354 | 129/132 |
| | | | RAM1 | 566T→A | L189Q | 133/137 |
| | | | SAN1 | 1464C→G | N488K | 97/100 |
| | | | SNF3 | 692C→A | A231D | 87/90 |
| EvoClone7C | 242 | 115 | GCR2 | 533T→A | L178Q | 148/154 |
| | | | IRA2 | 7049_7049delC | A2350Gfs2354 | 166/169 |

*Table 5. Continued on next page*

Table 5. Continued

| Strain name | Nominal generations | Number of mutations | Non-synonymous and promoter mutations segregating at evolved allele frequency >90% | Nucleotide change | Amino acid change | Mutant allele reads/total reads |
|---|---|---|---|---|---|---|
| | | | PDR1 | 2527 G→A | D843N | 145/159 |
| | | | PUF4 | 1960C→T | Q654* | 184/202 |
| EvoClone8 | 253 | 122 | ACE2 | −379 G→A | Promoter | 103/103 |
| | | | AXL2 | 700T→C | S234P | 87/87 |
| | | | ERG1 | 427 G→A | E143K | 160/163 |
| | | | HXK1 | 93A→T | E31D | 200/217 |
| | | | IFM1 | 1724T→A | I575N | 130/134 |
| | | | MIT1 | 188 G→A | W63* | 131/142 |
| | | | SKS1 | 1311C→G | Y437* | 152/152 |
| | | | SNF3 | 1237 G→A | E413K | 101/101 |
| | | | UBC5 | 443A→G | D118G | 137/141 |
| | | | UBR1 | 3859_3859delG | G1287Dfs1345 | 170/179 |
| EvoClone9 | 265 | 126 | ARE1 | −10 G→T | Promoter | 127/127 |
| | | | GCN2 | 4582A→C | I1528L | 125/137 |
| | | | GIN4 | 57 G→A | W19* | 79/81 |
| | | | IRC8 | 170 G→T | G57V | 118/119 |
| | | | MCD1 | 524C→T | S175L | 115/115 |
| | | | MCK1 | 675_675delG | G227Vfs249 | 114/116 |
| | | | MED1 | 1009C→G | L337V | 104/113 |
| | | | | 1465 G→T | E489* | 118/125 |
| | | | UBR1 | 3148_3148delC | L1050Yfs1063 | 141/145 |
| EvoClone10 | 242 | 196 | ACE2 | 1874A→T | Q625L | 121/122 |
| | | | AXL2 | 432_432delC | Y145Mfs154 | 161/164 |
| | | | BPH1 | 2369C→A | S790Y | 144/144 |
| | | | DNF2 | 2351T→C | F784S | 90/97 |
| | | | ECM5 | 3466 G→A | D1156N | 127/128 |
| | | | ENP2 | 1129T→A | F377I | 127/132 |
| | | | GAC1 | −7T→A | Promoter | 106/117 |
| | | | HTZ1 | −369T→C | Promoter | 69/73 |
| | | | KEM1 | 2268 G→A | M756I | 148/148 |
| | | | MCD1 | −28 G→T | Promoter | 153/153 |
| | | | MPT5 | 2409T→A | L590* | 146/146 |
| | | | MRPS17 | 325 G→A | D109N | 141/147 |
| | | | NUT1 | 2582C→A | S861* | 166/169 |
| | | | PRC1 | −283 G→A | Promoter | 138/148 |
| | | | RGT1 | 2060G→T | G687V | 91/91 |
| | | | SAC6 | 1736A→T | K542M | 125/137 |
| | | | TOP3 | 1679T→C | V560A | 100/103 |
| | | | UBR1 | 56T→A | L19Q | 130/140 |
| | | | WHI2 | 187 G→T | E63* | 138/139 |
| | | | WTM2 | −297T→A | Promoter | 125/129 |

Table 5. Continued on next page

*Table 5. Continued*

Nomenclature based on (**den Dunnen and Antonarakis 2000**): Mx→y: nucleotide change from x to y at base M, starting at base 1 (negative indicates promoter region). M_Ndelx: Deletion (Insertion:ins) of nucleotide x from base M to N. XNY: amino acid change from X to Y at codon N. * indicates stop codon. XNYfsN: as above, plus a frame shift mutation that results in stop codon at N.

The following mutations likely hitchhiked and were not included in this table: EvoClone3: *atg4* with *sin4*; EvoClone7A: *slx4* promoter with *ace2*; EvoClone7B: *thi3* with *ram1*, *crt10* with *ira2*, EvoClone8: *crh1* promoter with *ubr1*; EvoClone9: *brr1* with *MED1*, *nsp1* promoter with *irc8*; EvoClone10: *pri2* promoter with *rgt1*, *ino80* with *nut1*. This claim is based on the genetic linkage between the two alleles and the lower allele frequency of the mutation we argue is hitch-hiking relative to the putative causal mutation. The following mutations are not shown in the time courses in **Figure 6** and **Figure 6—supplements 1** and **2** because they were present at frequencies of less than 5% of the final population: EvoClone7C: *puf4-Q654**; EvoClone8: *mit1-W63**; EvoClone9 *mcd1-S175L*; EvoClone10: *top3-V560A* The following mutations were in the original, time zero strain and are not included in this table even through they segregated at > 90%: EvoClone2: *ira1-F664I*; EvoClone8: *phm8-I97N*, *rpl37a*(−52T→G), *vta1-A247V*, *yor1-E393D*; EvoClone9: *vta1-A247V*; EvoClone10: *aim32-E241G*, *irs4-N257S*, *nnt1*(−427T→C), *prp9-N155S*, *yrb1-N120I*.

mutated in four unrelated evolved clones (3 missense, 1 nonsense). Mutations that increase cAMP-dependent protein kinase activity are known to prevent starvation-induced cell cycle arrest (**Matsumoto et al., 1983**). By increasing Ras activity, loss of function mutations in *IRA1* and *IRA2* will elevate cAMP, which may allow cells to grow at lower external sugar concentrations than their ancestors.

Mediator is a multiprotein, global regulator of eukaryotic transcription. We found mutations in four of the at least 20 proteins that make up Mediator (**Myers and Kornberg, 2000**): *SIN4* (2 missense mutations), *CSE2* (1 nonsense mutation), *MED1* (1 nonsense mutation), and *NUT1* (1 nonsense mutation). Both mutations in *SIN4* occurred in residue 128 (G128D, G128S) and one, *sin4-G128D*, was also identified in a screen for suppressors of a deletion in *SWI6*. Swi6 associates with Swi4 to form SBF (SCB [Swi4-Swi6 cell cycle box] binding factor), a complex that regulates transcription early in the yeast cell cycle (**Li et al., 2005**). The *SIN4* mutations thus may cause increased expression of a gene normally activated by SBF. The hexose transporter gene, *HXT3*, (**Iyer et al., 2001**) is a known target of SBF and its was strongly elevated (108-fold and 32-fold) in the two clones (EvoClone1 and EvoClone4) carrying *SIN4* mutation (**Table 2 and 5**).

We analyzed read depth across all evolved clones and saw three major duplication or deletion events: (1) chromosome 3 in EvoClone10 was duplicated in the region between 152 and 172 kb from the left end of the chromosome. This doubling of read depth was reduced to a 50% increase in the segregated pool, indicating that the duplication was not causal. (2) The number of P-Type ATPases at the *ENO1/2/5* locus was reduced from 3 to 2 in EvoClone6 and EvoClone10. This reduction was also present in the backcrossed strain, indicating that it may have been causal. (3) The number of hexose transporter gene repeats at the *HXT6/7* locus increased from two in the ancestral strain to three in EvoClone7C. This amplification was also present in the backcrossed strain, indicating that it also may have been causal. Amplification of the *HXT6/7* locus has been found in other experimental evolutions (**Brown et al., 1998**; **Gresham et al., 2008**; **Kao and Sherlock, 2008**).

## Recreation of evolved clones verifies causal mutations

Are the putative causal mutations really responsible for the evolved phenotypes? We addressed this question by engineering the candidate mutations from two clones, EvoClone2 and EvoClone9, into their ancestor and asking if this manipulation reproduced the behavior of these clones. To recreate the sets of putative causal mutations, we replaced the five ancestral alleles in the ancestor with the five putative causal mutations in EvoClone2 to make Recreated2 and the eight putative causal mutations in EvoClone9 to make Recreated9. **Figure 7** and **Figure 3—figure supplement 2** shows that the morphology and the clump size distribution of the recreated strains are similar to that of the evolved clones.

To assess the fitness of the reconstructed strains, we competed them against their ancestor and the evolved clones. Both the evolved and recreated clones outcompeted their ancestor in 1–2 passages (**Table 1**) showing that they are much fitter on low sucrose. Both recreated strains were slightly less fit their evolved counterparts: there were fewer recreated than evolved cells after six passages (46 generations). This fitness defect has three possible causes: (1) the recreated clones are missing minor causal alleles, which we failed to find; (2) detrimental mutations were introduced during the

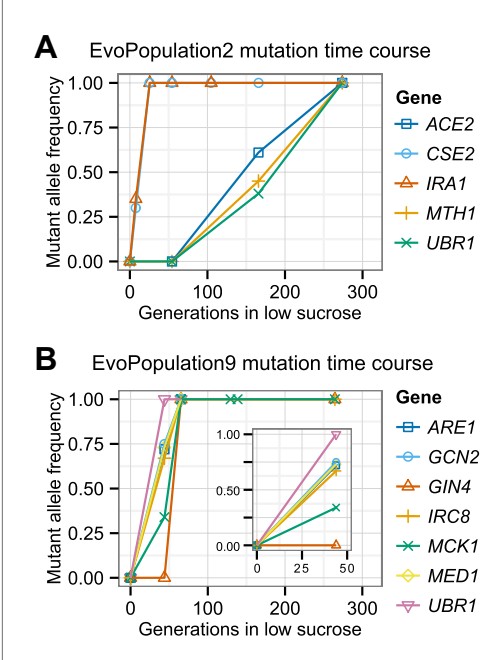

**Figure 6**. Putative causal mutation frequency at time points during the evolution. The alleles at the indicated time points were sequenced using Sanger sequencing, and frequencies were estimated from peaks in the trace plots. See *Figure 6—figure supplements 1* and *2* for the other evolved populations. See *Table 5* for amino acid and nucleotide changes.

The following figure supplements are available for figure 6:

**Figure supplement 1**. Putative causal mutation frequency at time points during the evolution for EvoClone 1, 3, 4, 5, 6, 7A, 7B, and 7C.

**Figure supplement 2**. Putative causal mutation frequency at time points during the evolution for EvoClone 8 and 10.

multiple transformations required to make the recreated strains; or, (3) the evolved clone (which is a mutator, unlike the recreated strain) continued to evolve and adapt during the competition.

To confirm that the putative causal alleles accounted for the evolved phenotype, we reverted these alleles in the evolved clones to their ancestral state. We replaced each of the mutant alleles in EvoClone2 and EvoClone9 with the ancestral allele. Both the resulting strains, Reverted2 and Reverted9, grew poorly in 1 mM sucrose, mostly existed as single cells, and were quickly outcompeted by the evolved clones (*Table 1*, *Figure 7*, and *Figure 3—figure supplement 2*).

We used bulk segregant analysis to ask whether individual alleles were required for the evolved phenotype. We crossed the recreated strains to their ancestors, isolated haploid spores, selected this population for growth on 1 mM sucrose, and measured the frequency of the evolved alleles in the selected population. All five mutations in Recreated2 were strongly selected for, six of the eight mutations in Recreated9 were strongly selected for, and one mutation in Recreated9 was moderately selected for, confirming that all but one of the thirteen putative causal mutations were causal for improved growth on low sucrose (*Figure 8*). We suspect that the eighth mutation in Recreated9, *gcn2-I1528L*, may be a false positive. In the original spore selection, the mutant allele segregated at only 91%, the low end of our threshold, and the mutation changed a poorly conserved isoleucine to leucine, suggesting it is unlikely to have a major effect on the protein's activity. In sum, our recreation experiments show that the putative causal alleles, with a single exception, are both necessary and sufficient to produce the evolved phenotype.

## Some causal mutations show antagonistic pleiotropy

We studied the role of the causal mutations in different growth conditions. Do the mutations selected on low sucrose increase, decrease, or have no effect on fitness on other carbon sources? To ask this question, we selected spores from crosses between ancestral and recreated clones in low monosaccharide and high glucose. Several mutations were selected for in low sucrose but not in low monosaccharide: mutations in *MTH1*, *CSE2*, and *ACE2* in Recreated2; and in *MCK1*, *IRC8*, and *GIN4* in Recreated9. The primary selective force for four of these six mutations is likely to be their ability to produce multicellularity: an *ace2Δ* strain is morphologically similar to EvoClone2 and a *mck1-G227Vfs249 irc8-G57V gin4-W19** strain is morphologically similar to EvoClone9 (*Figure 8*). Each of these two strains also outcompetes the ancestor strain in low sucrose but not in low monosaccharide (*Table 1*). This shows that multicellular clumps were specifically selected by growth in low sucrose, rather than being a general response to low sugar concentrations. But clumpiness alone was not sufficient to match the growth of the evolved strains in sucrose—the engineered clumpy strain (EngClumpy) was not as fit in 1 mM sucrose as one of the evolved clones that had similar clump size (EvoClone2) (*Table 1*).

Our evolved clones are less fit than their ancestor on high glucose (*Table 1*). There are two explanations for this observation. The first is a direct result of selection: some mutations that improve

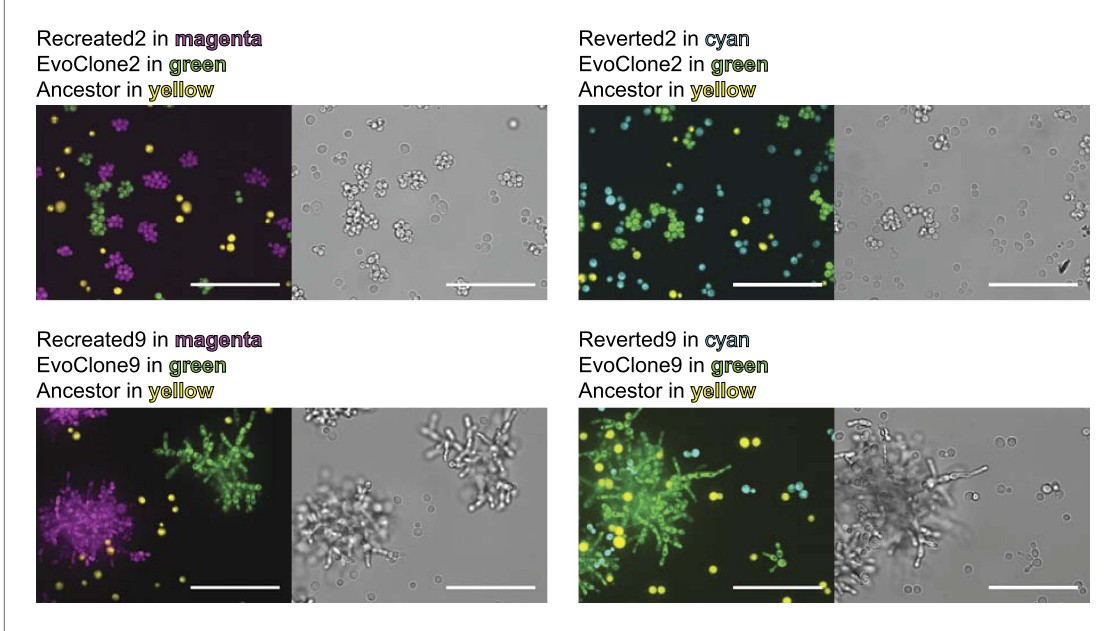

**Figure 7**. Engineering in alleles can recreate the evolved and ancestral morphologies. The ancestral strain was converted to the evolved morphology by converting ancestral alleles to those of the putative causal mutations and the evolved strains were converted to the ancestral morphology by converting the putative causal mutations to their ancestral alleles. The strains were grown separately in 1 mM sucrose and then mixed. The top row shows EvoClone2 strains and the bottom row shows EvoClone9 strains. The ancestor constitutively expresses mCherry and is shown in yellow; the evolved clone constitutively expresses YFP and is shown in green. The recreated strain (left) and the reverted strain (right) constitutively express CFP and are shown in magenta (recreated evolved) and cyan (reverted to ancestral). Representative confocal fluorescent (left) and brightfield (right) images were taken with a 60× objective in a glass-bottomed, 96-well plate. All scale bars are 50 μm.

growth on low sucrose reduce fitness on high glucose. The second appeals to the large number of mutations that hitchhiked with the causal mutations: some of the hitchhikers are neutral on low sucrose, but slow cell growth on high glucose. We distinguished these two possibilities by competing the evolved and recreated strains against their ancestor in high glucose. For both EvoClone2 and EvoClone9, the recreated strains performed as well as or better than the evolved clones, but worse than the ancestor, suggesting that both causal alleles and hitchhiking alleles contribute to the evolved clones' reduced fitness in high glucose.

We identified the causal alleles that impair growth on high glucose through bulk segregant analysis. When the progeny of the cross between ancestor and recreated strains are grown on high glucose, one of the causal mutations (*ubr1-C175Y*) in Recreated2 is selected against, three are approximately neutral, and one (*ira1-S2553C*) is strongly selected for. For Recreated 9, one mutation (*gin4-W19\**) is very strongly selected against, one (*ubr1-L1050Yfs1063*) is weakly selected against, three are roughly neutral, two are weakly selected for (*mck1-G227Vfs249* and *are1(-10 G→T)*), and one (*mcd1-S175L*) is very strongly selected for in high glucose (**Figure 8**). Note that the mutations in *UBR1* and *GIN4* have opposite effects in low monosaccharide: *ubr1* mutations are selected for in both low sucrose and low monosaccharide, whereas the *gin4-W19\** mutation is selected against in low monosaccharide.

To analyze the effect of individual mutations on expression of *HXT4* and *SUC2* in EvoClone2, we made seven strains, each with a different allelic combination of the five causal mutations. We then measured *HXT4* and *SUC2* expression using RT-qPCR in the seven new strains and in Recreated2. *Figure 8—figure supplement 1* shows the results. The increase in *HXT4* expression is clearly a result of the *MTH1* loss of function mutation (*mth1-H154Tfs156*): all combinations containing this allele have higher *HXT4* expression than those containing the ancestral *MTH1* allele. This result confirms the role of *MTH1* in hexose transporter repression. The gradient in *SUC2* expression across the allelic combinations indicates that *SUC2* expression is a complex phenotype, which is under the quantitative control alleles at several genes. We also checked the clumpiness of each strain by manual observation using a microscope: the strains with the wild type *ACE2* allele were not clumpy,

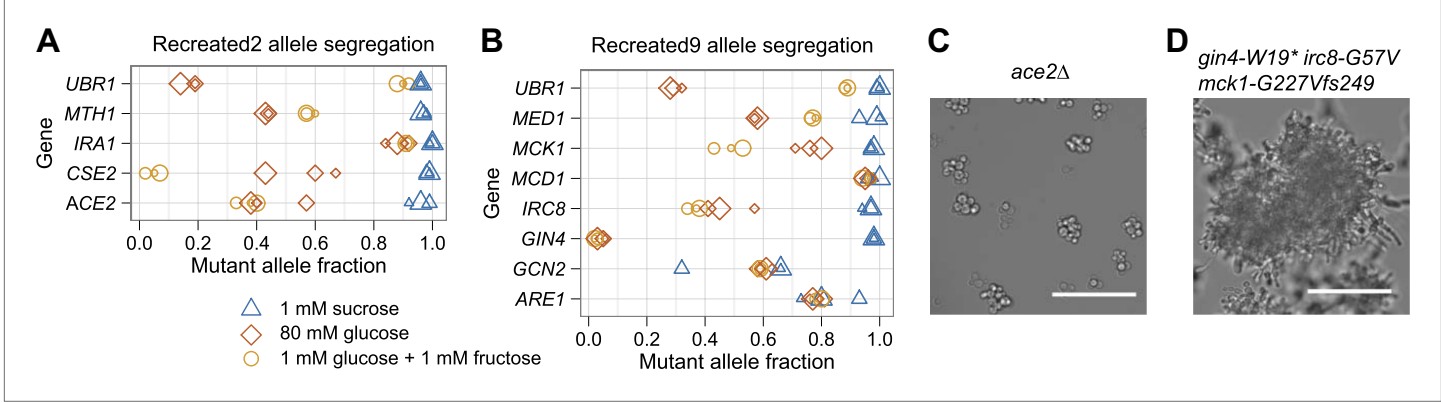

**Figure 8**. Bulk segregant analysis with the recreated strains verifies causal alleles and shows that alleles responsible for clumpiness are selected in low sucrose and not low monosaccharide. The recreated strains were backcrossed, sporulated, and selected in three different media: 1 mM sucrose (low sucrose), 80 mM glucose (high glucose), and 1 mM glucose plus 1 mM fructose (low monosaccharide). The mutant allele fraction was estimated from Sanger sequencing across the allelic variants. The size of the data point (small, medium, or large) for each allele and media combination refers to one of three independently derived diploids. (**A**) Recreated2 allele segregation. (**B**) Recreated9 allele segregation. (**C**) Ancestor strain with *ACE2* deleted (*ace2Δ*) has a clumpy phenotype. The *ACE2* mutation in Recreated2, a likely loss of function mutation that caused the clumpiness in EvoClone2, was selected for in low sucrose and was not selected for in low monosaccharide. (**D**) Ancestor strain that has wild type alleles of *IRC8*, *MCK1*, and *GIN4* replaced with the EvoClone9 alleles has a clumpy phenotype. All three mutant alleles were selected for in low sucrose and were not selected for in low monosaccharide. Representative DIC images were taken with a 40× objective in a glass-bottomed, 96-well plate. Scale bars are 50 µm.

The following figure supplements are available for figure 8:

**Figure supplement 1**. Change in *HXT4* and *SUC2* expression in various allelic combinations of Recreated2 compared to the ancestor.

while each strain with *ace2-L323** was clumpy. This result confirms that the clumpiness in EvoClone2 is caused by the loss of function mutation in *ACE2*.

## Discussion

In earlier work, we showed that clumps of yeast could grow in concentrations of sucrose where single cells could not. We speculated that the sharing of public goods (in this case, the hydrolysis products of sucrose) could select for multicellularity. Here, we confirmed this hypothesis: 11 of 12 clones evolved in low sucrose formed multicellular clumps caused by cells failing to separate after cell division (*Figure 3* and *Figure 3—figure supplements 1–3*). RNA sequencing revealed two additional strategies that most clones used: 10 elevated invertase expression, a strategy we had engineered, and 11 elevated hexose transporter expression, a strategy we had not engineered. We found no evidence of another strategy we had engineered: sucrose import. We cannot eliminate the possibility that the clones import some sucrose, but when we kept cells from secreting invertase, their fitness in low sucrose was severely reduced, showing that their growth still depends on invertase secretion. We identified the mutations that led to increased growth on low sucrose and showed that these were necessary and sufficient to recreate the evolved phenotype in two of the evolved clones. The genes that were mutated identified pathways that control cell growth and nutrient utilization.

The evolved cells clump because they fail to separate after cell division, rather than by flocculating (*Figure 4*). A loss of function mutation in *ACE2* was likely the primary contributor to the clumpy phenotype in at least 6 of the 11 multicellular clones. We were able to recreate the clumpy morphology of EvoClone9 with mutations in *MCK1*, *IRC8*, and *GIN4*; note that EvoClone3, one of the two other clumpy clones with wild type *ACE2*, also had a mutation in *IRC8*. Mutations in *ACE2* are likely to be frequent because inactivation of this gene simultaneously reduces the expression of multiple genes needed for cell separation; mutations in the genes that Ace2 regulates would be likely to have smaller effects. None of our clones became clumpy by restoring function to *AMN1*, the gene whose loss of function allele was selected for during laboratory domestication of budding yeast

(*Yvert et al., 2003*). This is not surprising since the target size for mutations that restore *AMN1* to its wild type function is much smaller than the target size for inactivating *ACE2*.

Was the clumpy phenotype selected because it confers some other advantage, such as faster settling in unstirred cultures (*Ratcliff et al., 2012*), rather than because it improves growth in low sucrose? Three lines of evidence show that this is not true. First, none of the evolved clones had a strong fitness advantage over the ancestor in 1 mM glucose plus 1 mM fructose, the monosaccharide equivalent of 1 mM sucrose (*Table 1*) showing that these clones were specifically adapted to sucrose. Second, a strain that was engineered to be clumpy (EngClumpy) outcompeted the ancestor in low sucrose, but not low monosaccharide (*Table 1*), demonstrating that low sucrose concentrations specifically selects for clumps, reinforcing our earlier findings on engineered multicellularity (*Koschwanez et al., 2011*). Third, the alleles that recreate the clumpy morphology (*ace2-L323\** in EvoClone2 and the combination of *mck1-G227Vfs249*, *gin4-W19\**, and *irc8-G57V* in EvoClone9) are strongly selected for in low sucrose and are not selected for in low monosaccharide (*Figure 8* and *Table 1*). Because the starting pool of segregants and the selection protocol was identical in both media, this result shows that more efficient sucrose utilization, rather than some other phenotype, selected for the clumpy phenotype.

Analyzing gene expression and the role of individual mutations demonstrated that we had selected for more than multicellularity. Most (10 out of 12) of the evolved clones had increased invertase expression, even though the strain we had engineered to have high invertase expression had only a small advantage over a wild-type strain (*Table 1*). The advantage of increased invertase expression in a shaken culture of single cells is likely to be small since all cells share the monosaccharides that escape from the small, unstirred volume that surrounds each cell. But high invertase expression makes sense in conjunction with the other two strategies we observed, multicellularity and elevated monosaccharide import. In a clump, multiple cells share a larger unstirred volume and have preferential access to the monosaccharides their neighbors release (*Koschwanez et al., 2011*). Increasing invertase expression will increase the rate of sucrose hydrolysis, and making more hexose transporters will increase the fraction of the monosaccharides that are imported rather than escaping to the bulk medium. Thus we expect that increasing invertase expression, making cell clumps, and making more hexose transporters will act together to increase fitness on low sucrose. Because low extracellular glucose induces the expression of both high affinity hexose transporters and invertase, it is likely that single mutations can increase levels of both types of proteins. In addition, clumpiness may indirectly elevate invertase expression: because secreted invertase expression peaks at roughly 0.5 mM external glucose (*Koschwanez et al., 2011*), clumps may increase invertase expression because concentrating the cells in space collectively raises the local glucose concentration.

Many of the mutations we selected are likely to affect the linked processes of sugar harvesting and the control of cell growth. As the level of potential nutrients in their environment falls, cells must make choices: how much of their resources they invest in trying to harvest nutrients from their environment, and how much of the imported nutrients they store to meet future challenges and how much they use to maximize their current growth rate. As an example, wild strains may stop growing and dividing at low external glucose levels because this decision improves their ability to survive if glucose levels continued to fall, even though they could support cell growth at the current glucose concentration if they expressed very high levels of hexose transporters and stored no sugar as glycogen. Influences on this type of decision are likely to explain repeated mutations in one gene (*UBR1*) and three groups of genes, which control glucose's effect on transcription (the *RGT1* group), how nutrient availability controls cell growth (the *IRA* group), and general transcription (the Mediator group).

*UBR1* was mutated in half of the evolved clones. Four of the six mutations are nonsense mutations, suggesting that we selected for inactivation of this protein. We suspect that prevention of N-end rule mediated protein degradation increases the expression of genes that promote growth in carbon limited cells: in both evolved clones that we recreated, the evolved allele of *UBR1* was selected for in sucrose and low monosaccharide and selected against in high glucose (*Figure 8*).

The *RGT1* group of genes gives external glucose concentrations the ability to control the expression of genes needed for growth in low glucose concentrations. Genes in this pathway were causally mutated in 8 of 12 evolved clones. The pattern of mutations we observe in these genes is consistent with selection to reduce the repression mediated by Rgt1 and Mth1. We also found, by using RT-qPCR on one of the reconstructed clones, that a loss-of-function mutation in *MTH1* was responsible for the increase in *HXT4* expression (*Figure 8—figure supplement 1*).

Mutations in one of the components of the Mediator complex, Sin4, had previously been shown to be involved in genes required for passage through Start, the transition in the yeast cell cycle that is most sensitive to nutritional signals. We recovered two mutations in this gene and found they were both associated with a strong increase in expression of Hxt3, a low affinity glucose transporter.

We saw mutations in *IRA1* and *IRA2* in four independent lineages. These mutations are likely to reduce the activity of Ira1 and Ira2 and thus increase the activity of Ras at low sugar concentrations, promoting growth. A mutant of *IRA1* in EvoClone2 was one of only two mutants of 13 in the recreated strains that was strongly selected for in all three conditions, and mutants of *IRA1* have been found in a yeast experimental evolution done in low glucose (*Kao and Sherlock, 2008*) and rich media (Lang, personal communication, October 2012). We suspect that Ira1 and Ira2 reduce the activity of Ras and thus the growth rate of cells on suboptimal carbon sources (low sugar concentrations, ethanol). Because we used batch cultures, even the high glucose cultures went through an environmental shift after they had fermented all the glucose to ethanol. Disruption of either *IRA1* or *IRA2* increases sensitivity to heat shock (*Tanaka et al., 1989*, *1990*). We suggest that we have selected for mutants than ignore pathways designed to induce cells exposed to unfavorable conditions to reduce their growth rate and induce programs that will protect them if conditions worsen.

We took an integrated approach to understanding evolution. Experimental evolution has long been used to study nutrient adaptation in yeast (*Paquin and Adams, 1983*; *Ferea et al., 1999*) and bacteria (*Levin et al., 1977*; *Lenski et al., 1991*; *Barrick et al., 2009*). Although evolved bacteriophage isolates were sequenced in the 1990s (*Bull et al., 1997*), the cost of cellular genome sequencing has only recently dropped to a level where it is possible to find single nucleotide polymorphisms in an evolved microbe (*Barrick et al., 2009*; *Araya et al., 2010*). We combined engineering, experimental evolution, bulk segregant analysis, and genetic reconstruction to build a picture of how budding yeast evolve in response to a specific challenge: the need to grow on a nutrient that is released into the local environment by extracellular hydrolysis. Testing hypothetical strategies by engineering allowed us to look for their phenotypic signatures, such as the increased expression of hydrolytic enzymes, in evolved populations. Having a method to sift the minority of causal mutations from the majority of hitchhikers allowed us to use mutators and increase the speed of evolution. The ability to swap evolved and ancestral alleles made it possible to verify our methods for identifying causal mutations and show that we had identified mutations that were either generalists, promoting growth across all the environments we tested, or specialists showing antagonistic pleiotropy by being strongly beneficial in one environment and deleterious in another. Finally, performing multiple experiments in parallel allowed us to identify mutations in particular genes and pathways as contributing strongly to evolutionary success.

We speculate that unicellular organisms have evolved a combination of three strategies to access nutrients that are digested extracellularly: increasing secretion of an extracellular hydrolase, increasing the import of its products, and forming multicellular clumps. An increase in nutrient transport seems the least costly, but its effect as a sole strategy is limited if the local concentration of the hydrolysis products remains unchanged. An increase in enzyme secretion will increase the level of the hydrolysis products, but enzyme production is costly and allows unrelated cells from the same or different species to 'cheat' and steal simple nutrients without incurring the cost. The formation of multicellular clumps allows the hydrolysis products to be shared among direct kin and reduces the risk of cheating (*Koschwanez et al., 2011*). We have shown that yeast, when selected for growth in low sucrose media, used a mixture of all three strategies.

The combination of laboratory engineering and evolutionary analysis shown here and in our previous work supports the speculation that the sharing of public goods was the initial selection for undifferentiated multicellularity. For organisms with rigid cell walls like yeast, algae, and bacteria, cells must secrete enzymes to separate mother from daughter after cytokinesis. The target size for mutations in these enzymes and the control circuits that regulate their production will be large, making it easy to evolve undifferentiated multicellularity. Although lab strains of yeast were selected to be unicellular during their domestication (*Mortimer and Johnston, 1986*), the level of clumpiness varies in wild isolates of yeast, and undifferentiated multicellularity may have been recurrently selected for its advantage in harvesting nutrients or for protection from stress (*Smukalla et al., 2008*). Organisms without a cell wall, such as animal cells, would need another innovation, such as cell-adhering cadherins (*Abedin and King, 2008*), in order to evolve multicellular clumps. But even in these organisms, cells must adhere to each other during sexual reproduction, implying the existence of adhesive proteins that could be modified to allow the evolution of multicellular clumps.

If undifferentiated multicellularity is so easy to evolve in eukaryotes, then why did it take so long for fossils of differentiated multicellular organisms to appear in the fossil record? Indeed, there is suggestive evidence of undifferentiated eukaryotic multicellularity 2 billion years ago (*El Albani et al. 2010*), but the radiation of more complex multicellularity is not seen for over a billion years later. This disparity has received a large amount of attention in paleontology, particularly in the context of the seemingly rapid radiation of the animals at the base of the Cambrian. Possible explanations for this 'Cambrian explosion' include novel selective forces due to the environmental changes that occurred shortly before the Cambrian or the possibility that evolutionary arms races among a few early species drove the radiation (*Marshall, 2006*; *Peters and Gaines, 2012*). Alternatively, the evolution of differentiated multicellularity may have been slow and gradual and we simply lack the fossil evidence. Whatever the case, it is interesting to ask why the first differentiation might have arisen. From our data, we speculate that the effectiveness of clumping as a strategy could be improved if a limited number of cells increase their enzyme secretion. Such a simple division of labor may have been the precursor to complex multicellularity in the lineages that lead to animals and other complex multicellular species.

## Materials and methods

### Media

Unless otherwise noted, the experimental evolution and all experiments were conducted in minimal (no amino acids or nucleotides) synthetic media with the following two exceptions: (1) the auxotrophic strains expressing high levels of Flo1 required the addition of leucine, histidine, and lysine, and (2) strains undergoing galactose induction in *Figure 2* were pregrown in YP 2% glycerol (10 g/l yeast extract, 20 g/l peptone, and 2% [v/v] glycerol) plus the indicated concentration of galactose. The minimal synthetic media was made with refrigerated sugar stocks and refrigerated 10× yeast nitrogen base (YNB) that was based on Wickerman's recipe (*Wickerham, 1951*) but without riboflavin, folic acid, and inositol (see [*Koschwanez et al., 2011*] for the recipe). 200 units/ml Penicillin and 0.2 mg/ml streptavidin were added to the experimental evolution media to prevent bacterial contamination. Unless otherwise noted, all chemicals used in this research were purchased from Sigma-Aldrich (www.sigmasldrich.com).

### Experimental evolution

Each of the 10 evolved cultures was derived from a haploid *MAT**a*** W303 prototrophic strain that expressed a yellow fluorescent protein (YFP) variant from the constitutive *ACT1* promoter. See *supplementary file 3* for a description of the starting strains and their construction. To start the evolution, a single colony from each strain was inoculated into 80 mM glucose media and grown for 7–20 generations, and $5×10^5$ cells were then inoculated into a 125 ml flask containing 50 ml of 1 mM sucrose media. The flasks were placed at 30°C on a shaker rotating at 120 rpm. When there was visible growth in a culture (>$10^6$ cells/ml), it was passaged as follows. First, the culture was spun down and resuspended in 1 ml of yeast nitrogen base (YNB) to ensure that non-sucrose carbon sources, created by invertase activity or cellular metabolism, were not inoculated into the new cultures. Second, a sample of resuspended cells were mixed with glycerol and frozen at −80°C. Third, the cell density was measured with a Coulter counter (www.beckmancoultermedia.com). Finally, the resuspended cells were vortexed and $5×10^5$ cells were inoculated into a new 125 ml flask containing 50 ml of warm 1 mM sucrose media. When the cells became too clumpy to count accurately in the Coulter counter, the cultures were instead diluted at a 500:1 ratio into fresh media.

### Clonal selection

To pick clones, eight replicates of each culture were serial diluted 2:1 twelve times in 1 mM sucrose media in 96-well plates such that the last dilution contained an estimated average of less than one cell per well for each replicate. For each population, there were thus eight parallel sets of dilutions. The plates were incubated on a plate shaker at 30°C for 3 days, and each well was examined using a microscope for growth in sucrose and for morphology. A well was assumed to contain a clonal population at the lowest starting cell density where growth was seen if there was a lower density where growth was not seen in any of the eight wells. Each of the selected clones was verified to grow from low density in 1 mM sucrose within 2 days.

### Bulk segregant analysis

*URA3* was deleted from each YFP-labeled *MAT**a*** evolved clone. The *ura3Δ* strain was mated with yJHK519, a *MATα, trp1Δ::kanMX4 ura3::P_{STE2}-URA3* derivative of the ancestor labeled with CFP. In this strain, the

endogenous *URA3* promoter is replaced with the *STE2* promoter, which is only induced in *MAT*a cells, making it possible to select for *MAT*a spores after meiosis. Mating was performed by mixing cells from the two strains together on a YPD plate with a toothpick and growing overnight at 30°C. The mating mixtures were then plated onto G418, -TRP plates, and a diploid strain was selected from a colony on the plate.

To sporulate the diploid strains, cultures were grown to saturation in YPD, and then diluted 1:50 into YP 2% acetate. The cells were grown in acetate for 12–24 hr and then pelleted and resuspended in 2% acetate. After 4–5 days of incubation on a roller drum at 25°C, sporulation was verified by observing the cells in a microscope.

To digest ascii, 1 ml of the sporulated culture was pelleted and resuspended in 50 µl 10% Zymolyase (www.zymoresearch.com) for 1 hr at 30°C. 400 µl of water and 50 µl of 10% Triton X-100 were added, and the digested spores were sonicated for 5–10 s to separate the tetrads. Tetrad separation was verified by observing the cells in a microscope. The separated tetrads were then spun down slowly (6000 rpm) and resuspended in growth media.

To select haploid spores, the entire digested spore culture was added to 50 ml of 1 mM glucose + 1 mM fructose minimal media and growth to saturation. This media selected for *TRP1,* haploid *MAT*a cells: neither haploid *MAT*α nor diploid *MAT*a/*MAT*α cells can express *URA3* from the *STE2* promoter. Each culture was selected for growth in 1 mM sucrose through four passages (inoculation to saturated growth) as follows: one 100:1 dilution, and three 500:1 dilutions (≈34 generations total). The cells were spun down and resuspended in YNB during each dilution in order to eliminate the non-sucrose carbon sources from the media. Genomic DNA was made from the final saturated culture.

Bulk segregant analysis of Recreated2 and Recreated9 followed the same procedure with one exception: the cells were not spun down in between dilutions. Instead, 100 µl of culture was added directly to 50 ml of fresh media for a 500:1 dilution.

## Genomic DNA preparation

To prepare genomic DNA, the culture was pelleted and resuspended in 50 µl of 1% Zymolyase in 0.1 M, pH 8.0 NaEDTA. The cells were incubated for 30 min at 37°C to digest the cell wall, and then the cells were lysed by adding 50 µl 0.2 M NaEDTA, 0.4 M pH 8.0 Tris, 2% SDS and incubated at 65°C for 30 min. 63 µl of 5 M potassium acetate was added, and the mixture was incubated for 30 min on ice. The insoluble residue was then pelleted, and 750 µl of ice-cold ethanol was added to 300 µl of the supernatant to precipitate the DNA. The DNA was pelleted, and the pellet was resuspended in 0.2 mg/ml RNAase A. After 1 hr of incubation at 37°C, 2 µl of 20 mg/ml Proteinase K was added, and the solution was incubated for an additional 2 hr at 37°C. The DNA was again precipitated by adding 130 µl isopropanol. The DNA was pelleted, briefly washed with 70% ethanol, repelleted, and resuspended in 100 µl 10 mM Tris, pH 8.0.

## RNA isolation

To isolate RNA, cells in log-phase growth were fixed by adding 6 ml of culture to 9 ml of ice-cold methanol and then incubating at −20°C for 10 min. Cells were pelleted at 4°C, resuspended in 1 ml of RNAase-free ice-cold water in a 2 ml cryogenic storage vial, and then repelleted at 4°C. RNA was then isolated with acidic phenol using a published protocol (*Collart and Oliviero, 2001*).

## RT-qPCR

Isolated RNA was treated with DNase I (Thermo Scientific EN0525, www.thermofisher.com) according to manufacturer's instructions. Intact RNA was verified by observing two sharp rRNA bands using agarose gel electrophoresis. cDNA was made from 100–200 ng of RNA using Thermo Scientific Maxima First Strand cDNA Synthesis Kit for RT-qPCR (K1641). The 20 µl reaction was diluted tenfold and 11 µl of the diluted sample was used for real time qPCR using Thermo Scientific Maxima SYBR Green/ROX qPCR Master Mix (K0222). Amplification efficiency of the primers was verified by generating standard curves with four serial dilutions. The following primers were used: *SUC2* (GCTTTCCTTTTC-CTTTTGGCTGG, TCATTCATCCAGCCCTTGTTGG); *HXT4* (TTGGGTTACTGTACAAACTACG, TGTCA-TACCACCAATCATAAAC); *ALG9* (GTTTAATCCGGGCTGGTTCCAT, TAGACCCAGTGGACAGATAGCG). *ALG9* was used for normalization (*Teste et al., 2009*). The $\Delta\Delta C_T$ method was used to find change in RNA expression (*Livak and Schmittgen, 2001*). Three independent trials were performed for each reported RT-qPCR result and the data reported is the mean difference in expression between two strains.

## Sequencing

DNA and RNA libraries were prepared for sequencing using the Illumina TruSEQ kit (www.illumina.com) and were sequenced on an Illumina HiSeq 2000. Mean coverage across the genome was as follows: ancestor DNA—56×, evolved clone DNA—10× minimum, sporulated pools—90× minimum, RNA—40× minimum. Single end, 50 bp reads were used for the ancestor DNA, evolved clone DNA, and RNA. Paired end, 100 bp reads were used for the sporulated pool DNA. Sequencing data is deposited in the Sequence Read Archive.

## Sequence analysis

DNA sequences were aligned to the S288C reference genome r64 (downloaded from the Saccharomyces Genome Database, www.yeastgenome.org) using the Burrows-Wheeler Aligner (bio-bwa.sourceforge.net) (*Li and Durbin, 2009*). The resulting SAM (Sequence Alignment/Map) file was converted to a BAM (binary SAM) file, sorted, indexed, and made into a pileup format file using the samtools software (samtools.sourceforge.net) (*Li et al., 2009*). Indels were realigned locally using GATK (www.broadinstitute.org/gatk/) (*McKenna et al., 2010*), and variants were called from the pileup file using the Varscan software (varscan.sourceforge.net) (*Koboldt et al., 2012*).

To perform the segregation analysis, we wrote a custom sequencing pipeline in Python (www.python.org), using the Biopython (biopython.org) and pysam (code.google.com/p/pysam/) modules, that finds sequence variants between the ancestor and clone, classifies each variant as a nonsynonymous coding region, synonymous coding region, or promoter mutation, and ranks each mutation by its segregation frequency. All software written for this analysis is publicly available at https://github.com/koschwanez. We used the following criteria to select putative causal mutations: (1) the evolved clone was mutated relative to the strain used to inoculate the time zero culture, (2) the mutation caused a nonsynonymous substitution in a coding region or changed the promoter sequence ≤500 bp upstream of the coding region start site, and (3) the mutation appeared in over 90% of the reads in backcrossed pool. Putative causal mutations were manually verified by looking at aligned reads.

RNA sequences were aligned to the S288C reference genome using TopHat (*Trapnell et al., 2009*), and significant differences in expression between the ancestor and the evolved clone were called using the default setting in Cufflinks (*Trapnell et al., 2010*). The Cufflinks package uses the log of the ratio of expression in two conditions together with an estimate of the gene's variance to generate a t-value that it uses in a Students t-test. We used one ancestor derivative, yJHK110, as the reference strain to compare expression with the evolved clones, and the other ancestor derivative, yJHK111, as a control strain in the comparison.

## Data analysis and figures

Unless otherwise noted, data analysis was performed in the R programming language (www.r-project.org) and plots were generated using the R library ggplot2 (*Wickham, 2009*). The Adjusted Wald method of calculating 95% binomial confidence intervals (*Agresti and Coull, 1998*) was used in *Figure 2* because a low number of samples were used to generate a binomial mean. Cartoons were made using Adobe Illustrator CS5 (www.adobe.com).

## Sanger trace plot analysis

Commercial Sanger sequencing returns trace plots, chromatograms that indicate the relative frequency of each base at each position in the sequenced DNA. Trace plots were used in two analyses. First, we estimated the spread of mutations through a population by Sanger sequencing time points from the evolution. Second, we estimated the segregation of alleles in the backcrossed, recreated strains by Sanger sequencing the putative causal alleles in the final, selected population. The fraction of mutant alleles in the population was assumed to be the height of the mutant allele peak divided by the height of the mutant allele peak plus the ancestor allele peak. In the time course analysis, values below 5% (the approximate background level) were assumed to be zero, and values above 95% were assumed to be 100%. In the segregation analysis, actual values, not corrected for background, are shown.

We note that many of the mutations appear to rise with very similar time courses to each other. This could reflect the modest time resolution of our measurements and the insensitivity of using Sanger sequencing to estimate allele frequencies, but we suspect it reflects the strong advantages that accrue to those rare lineages where two or three beneficial mutations occur in quick succession (*Lang et al., 2011*).

## Competitions

To compare the fitness of the recreated and evolved strains, we developed a fitness assay that was suitable for strains with non-uniform morphologies. We could not use standard, quantitative methods of measuring fitness, such as FACS-based competitions (*Desai et al., 2007*), because we were not able to accurately count the number of cells per clump in a large population. We therefore used a qualitative microscopy-based fitness assay to compare growth of a YFP-labeled population and a CFP-labeled population. We started the competitions by growing each population separately to saturation in 1 mM glucose + 1 mM fructose. Equal volumes of each strain were mixed together, and then checked under a microscope to verify that both strains were equally present. 50 µl of the mixed culture was then inoculated into 10 ml of the test media. When the culture reached visible density (>10$^6$ cells/ml), three blindly chosen images of a 20 µl sample were taken on the microscope with a 20× objective. Extermination was defined as zero cells in one strain and more than 30 total cells of the other strain. If both strains were still present, 50 µl of the culture was inoculated into 10 ml of fresh media. After six growth cycles, a winner was declared if there was a clear majority (>75%) of one strain. Otherwise, the competition was declared a tie. The number of cycles until extermination was averaged across three independent experiments, and qualitative values (++++ through −−−−) were assigned as described in *Table 1*. + or − was assigned if one of the strains won two out of the three independent competitions that lasted through all six growth cycles.

## Check for flocculation or cell separation

The evolved clones already constitutively expressed YFP, and a constitutively expressed CFP version of each clone was made by deleting the P$_{ACT1}$-ymCitrine-*URA3* construct from the *URA3* locus and then transforming with a *ura3Δ*::P$_{ACT1}$-yCerulean-*URA3* construct. The two versions of each strain were grown separately in 1 mM glucose + 1 mM fructose media to saturation, and then the two versions were vortexed or sonicated (both gave identical results) for 10 s and then combined at low density in fresh 1 mM sucrose media. The mixed cultures were grown to visible density at 30°C and then pipetted into a well of a glass-bottomed 96-well plate and examined by microscope. In each of the three independent experiments, at least 50 clumps were checked and three representative images were taken for each strain.

## Microscopy

Images except those in *Figure 7* were taken in a 96-well glass-bottomed plate (greiner bio-one, www.gbo.com) on a Nikon Ti inverted microscope (www.nikoninstruments.com) with a Photometrics CoolSnap HQ camera (www.photometrics.com) and MetaMorph software (www.metamorph.com). *Figure 7* images were taken on a Nikon Ti inverted microscope with a Yokogawa spinning disc confocal unit (www.yokogawa.com), 447, 515, and 594 nm lasers (www.spectraloptics.com), a Hamamatsu Orca camera (www.hamamatsu.com), and MetaMorph software. Images were converted to 8-bit, projected, adjusted for contrast, and annotated with scale bars using the Fiji distribution of ImageJ (*Schindelin et al., 2012*). Contrast was changed for visibility only; our results do not depend on illumination levels and are not affected by the change in contrast.

## Strain reconstruction and reversion

Alleles were replaced in both the evolved and ancestor strains by transforming with a *URA3* plasmid that contained a portion of the targeted gene with the desired mutation and was digested at a single cut site within the target gene (*Rothstein, 1991*). See *Figure 5—figure supplement 1* for the complete protocol. Transformants were selected on—URA plates, colonies that grew were streaked out on—URA plates, and the target sequence was amplified and sequenced to verify insertion. The colony was grown overnight in YPD (10 g/l yeast extract, 20 g/l peptone, and 2% [w/v] dextrose) and selected on 5FOA plates to identify cells in which the *URA3* construct had looped out (*Boeke et al., 1984*). Colonies that grew were streaked out on 5FOA, replica plated to YP 2% Acetate (20 g/l agar, 10 g/l yeast extract, 20 g/l peptone, and 2% [w/v] potassium acetate) to eliminate petites, and the target region was amplified and sequenced to ensure that only the desired allele was present. Restriction enzymes, DNA polymerase, polynucleotide kinase, and ligation enzymes were purchased from New England Biolabs (www.neb.com).

## Acknowledgements

The authors thank Erin O'Shea, Vlad Denic, Mike Laub, Greg Lang, Jim Bull, Rama Ranganathan, and members of the Murray Lab for reviewing the manuscript.

# Additional information

## Funding

| Funder | Grant reference number | Author |
|---|---|---|
| National Institute of General Medical Science | K25GM085806 | John H Koschwanez |
| National Institute of General Medical Science | GM06873 | John H Koschwanez, Kevin R Foster, Andrew W Murray |

The funders had no role in study design, data collection and interpretation, or the decision to submit the work for publication.

## Author contributions

JHK, Conception and design, Acquisition of data, Analysis and interpretation of data, Drafting or revising the article; KRF, Conception and design, Drafting or revising the article; AWM, Conception and design, Analysis and interpretation of data, Drafting or revising the article

# Additional files

## Supplementary files

- Supplementary file 1. (**A**) Genes whose expression changed significantly in nine or more evolved clones. (**B**) Genes whose expression was increased or reduced at least tenfold. (**C**) All genes whose expression changed significantly in any clone or the control strain. Values are $\log_2$ change from ancestor to sample strain. Genes whose value did not change significantly are shown as zero.

- Supplementary file 2. Putative causal mutation pathway summary.

- Supplementary file 3. Strain list.

## Major datasets

The following dataset was generated:

| Author(s) | Year | Dataset title | Dataset ID and/or URL | Database, license, and accessibility information |
|---|---|---|---|---|
| Foster K, Murray AW, Koschwanez JH | 2013 | Sequences for: Improved use of a public good selects for the evolution of undifferentiated multicellularity | PRJNA192270; http://www.ncbi.nlm.nih.gov/bioproject/192270 | Publicly available at the Sequence Read Archive (http://www.ncbi.nlm.nih.gov/sra). |

The following previously published dataset was used:

| Author(s) | Year | Dataset title | Dataset ID and/or URL | Database, license, and accessibility information |
|---|---|---|---|---|
| Cherry JM, Hong EL, Amundsen C, Balakrishnan R, Binkley G, Chan ET, Christie KR, Costanzo MC, Dwight SS, Engel SR, Fisk DG, Hirschman JE, Hitz BC, Karra K, Krieger CJ, Miyasato SR, Nash RS, Park J, Skrzypek MS, Simison M, Weng S, Wong ED | 2012 | Saccharomyces Genome Database: the genomics resource of budding yeast | R64-1-1; http://downloads.yeastgenome.org/sequence/S288C_reference/genome_releases/ | Publicly available at http://www.yeastgenome.org/. |

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
