## [Decision Letter]

Thank you for choosing to send your work entitled “Improved use of a public good selects for the evolution of undifferentiated multicellularity” for consideration at *eLife*. Your article has been favorably evaluated by a Senior editor and 2 reviewers, one of whom is a member of our Board of Reviewing Editors.

The Reviewing editor and the other reviewer discussed their comments before we reached this decision, and the Reviewing editor has assembled the following comments to help you prepare a revised submission.

In this paper the authors have set out to test the hypothesis that multicellularity can emerge in otherwise single-celled microbes by selection for improved growth in low-sucrose environments. Sucrose utilization in budding yeast is unusual as it must be hydrolyzed extracellularly by invertase, resulting in glucose and fructose, which can then be imported into the cell. In low sucrose conditions, cells can't grow when they exist as single cells as hydrolyzed glucose and fructose is lost through diffusion. However, the authors have previously shown that if cells are engineered to clump together they can grow in this condition. This is presumably because the local concentration of the products of sucrose hydrolysis (glucose and fructose) is sufficiently high for cells to grow when they clump but not when they are present as individual cells. In this paper the authors have performed a long-term selection in which wild type, non-clumpy cells were propagated by serial dilution in an environment containing low sucrose. They find that this condition does indeed result in selection for clumpy cells. In addition, most of the clones that they analyze have increased expression of the invertase gene (SUC2) and, surprisingly, increased expression of a large number of hexose transporter genes. The authors used a mutator strain and thus mutants with increased fitness have acquired a large number of mutations. To identify causative alleles the authors performed bulk segregant analysis, which allowed them to identify the likely alleles that confer increased fitness. They prove that a fitness increase is attributable to some of these mutations by engineering strains containing only these mutations in addition to reverting acquired mutations in the evolved strains. They speculate that clumping of unicellular organisms may have led to the development of multicellularity.

This is an excellent dataset that should definitively be published in a major journal. In fact, the experiments, as well as the analysis and the presentation, are mostly very clear. Still, there are some minor comments that need attention in a revised version.

1) There is little discussion about the general implications of the findings. Hence, the reader is left with the impression that it would have been rather easy to evolve the first steps to multicellularity, but would ask him or herself why it took such a long time until the first multicellular organisms emerged. Maybe the system studied is actually an oversimplification that has only an indirect implication for the evolution of multicellularity. We are of course aware that any such discussion would be rather speculative, but it might nonetheless be worthwhile to address at least some of the pertinent issues.

2) Regrettably, the authors have neglected to include an essential control: a population of cells propagated in parallel in a non-sucrose environment. Ideally, this would be a low glucose/fructose environment using identical methods. Using long term-selections it is very easy to select for clumping cells either intentionally (Ratcliff et al 2012) or unintentionally (Lang et al., 2011). As the authors set out to demonstrate that it is a benefit of sharing secreted invertase that drives selection for clumpiness, we would have expected this simple control. Alternatively, the same selection using a cytoplasmically expressed invertase and a *MAL11* overexpression allele would also serve as a negative control. This would provide the proof that the observed outcome is unique to the sucrose selection. Nevertheless, we do not consider the lack of this control as a fatal flaw as the authors present multiple post hoc analyses to demonstrate that the fitness increase is unique to a low sucrose environment. The most compelling being the demonstration that the evolved clones have no fitness difference (or slightly reduced fitness) compared with wild type in a low glucose/fructose environment.

3) The authors use of bulk segregant analysis to identify beneficial mutations and engineering to recreate strains is impressive. However, some simple tetrad dissection would enable the authors to make more definitive statements about the effect of ace2 mutations on clumpiness and Rgt1 pathway mutations' HXT expression (assayed using qPCR). In addition they could test whether the Ubr1 mutations segregate with any of the phenotypes. Segregation analysis of *SUC2* expression in a cross would test whether it is always associated with clumpiness.

4) The fact that the “recreated” strains are less fit than the evolved clones suggests that something is being missed. One obvious explanation is copy number variation. Studies from the Rosenzweig, Botstein, and Sherlock labs have shown that long-term propagation of yeast strains in nutrient poor environments using chemostats results in selection for transporter amplifications. As the low sucrose environment is also a nutrient poor one, we would expect amplification alleles to be beneficial. However, the authors' data do not allow testing for this as they have performed RNASeq analysis of clones. Were any individual clones sequenced that can be analyzed for copy number variation? The authors could also check in the bulk segregant sequencing data for evidence of copy number variation at the *SUC2* locus or any of the HXT loci that appear to be massively increased in expression.

5) Many wild *S. cerevisiae* strains are clumpy and loss of clumpiness is likely a lab-adapted trait. The authors should make it clear that loss and gain of clumpiness is likely to have occurred many times in yeast populations and provide other explanations for why clumpiness in yeast cells might be recurrently selected.

---

## [Author Response]

*1) There is little discussion about the general implications of the findings. Hence, the reader is left with the impression that it would have been rather easy to evolve the first steps to multicellularity, but would ask him or herself why it took such a long time until the first multicellular organisms emerged. Maybe the system studied is actually an oversimplification that has only an indirect implication for the evolution of multicellularity. We are of course aware that any such discussion would be rather speculative, but it might nonetheless be worthwhile to address at least some of the pertinent issues*.

We added the following as the last paragraph of the Discussion section:

“If undifferentiated multicellularity is so easy to evolve in eukaryotes, then why did it take so long for fossils of differentiated multicellular organisms to appear in the fossil record? Indeed, there is suggestive evidence of undifferentiated eukaryotic multicellularity 2 billion years ago (El Albani et al. 2010), but the radiation of more complex multicellularity is not seen for over a billion years later. This disparity has received a large amount of attention in paleontology, particularly in the context of the seemingly rapid radiation of the animals at the base of the Cambrian. Possible explanations for this “Cambrian explosion” include novel selective forces due to the environmental changes that occurred shortly before the Cambrian or the possibility that evolutionary arms races among a few early species drove the radiation (Marshall 2006; Peters and Gaines 2012). Alternatively, the evolution of differentiated multicellularity may have been slow and gradual and we simply lack the fossil evidence. Whatever the case, it is interesting to ask why the first differentiation might have arisen. From our data, we speculate that the effectiveness of clumping as a strategy could be improved if a limited number of cells increase their enzyme secretion. Such a simple division of labor may have been the precursor to complex multicellularity in the lineages that lead to animals and other complex multicellular species.”

*2) Regrettably, the authors have neglected to include an essential control: a population of cells propagated in parallel in a non-sucrose environment. Ideally, this would be a low glucose/fructose environment using identical methods. Using long term-selections it is very easy to select for clumping cells either intentionally (Ratcliff et al 2012) or unintentionally (Lang et al., 2011). As the authors set out to demonstrate that it is a benefit of sharing secreted invertase that drives selection for clumpiness, we would have expected this simple control. Alternatively, the same selection using a cytoplasmically expressed invertase and a* MAL11 *overexpression allele would also serve as a negative control. This would provide the proof that the observed outcome is unique to the sucrose selection. Nevertheless, we do not consider the lack of this control as a fatal flaw as the authors present multiple post hoc analyses to demonstrate that the fitness increase is unique to a low sucrose environment. The most compelling being the demonstration that the evolved clones have no fitness difference (or slightly reduced fitness) compared with wild type in a low glucose/fructose environment*.

We are frequently asked this question and the simplest answer would indeed be to be able to state that we had done the experiment suggested by the reviewer and found no multicellularity in strains evolved on low monosaccharide concentrations. There are three strong lines of evidence that argue that the evolution of multicellularity was a specific response to low sucrose concentrations: (1) the failure of the evolved clones to show increased fitness on low monosaccharide concentrations; (2) the fitness advantage of engineered clumpy strains in sucrose and not in low monosaccharide; (3) the failure to select for the mutations that cause multicellularity (*ace2* in Recreated 2 and *mck1*, *irc8*, and *gin8*) on low monosaccharide concentrations when we analyze the effect of individual mutations as they segregate in a cross between the ancestral and evolved strains.

*3) The authors use of bulk segregant analysis to identify beneficial mutations and engineering to recreate strains is impressive. However, some simple tetrad dissection would enable the authors to make more definitive statements about the effect of ace2 mutations on clumpiness and Rgt1 pathway mutations' HXT expression (assayed using qPCR). In addition they could test whether the Ubr1 mutations segregate with any of the phenotypes. Segregation analysis of* SUC2 *expression in a cross would test whether it is always associated with clumpiness*.

We made 9 allelic combinations of strain Recreated2 and tested them for clumpiness, *HXT4* expression, and *SUC2* expression using qPCR. We concluded that the *ACE2* mutation caused clumpiness and the *MTH1* mutation caused increase in *HXT4* expression (as the reviewers suspected). The *SUC2* expression was more complex and we were not able to make any conclusions from these data. The *UBR1* mutation did not seem to influence any of the phenotypes. These additional experiments are detailed in the Results and the Materials and methods sections. The data from the experiment is in Figure 8–figure supplement 1.

*4) The fact that the “recreated” strains are less fit than the evolved clones suggests that something is being missed. One obvious explanation is copy number variation. Studies from the Rosenzweig, Botstein, and Sherlock labs have shown that long-term propagation of yeast strains in nutrient poor environments using chemostats results in selection for transporter amplifications. As the low sucrose environment is also a nutrient poor one, we would expect amplification alleles to be beneficial. However, the authors' data do not allow testing for this as they have performed RNASeq analysis of clones. Were any individual clones sequenced that* can *be analyzed for copy number variation? The authors could also check in the bulk segregant sequencing data for evidence of copy number variation at the* SUC2 *locus or any of the HXT loci that appear to be massively increased in expression*.

We had looked for evidence of copy number variation across all genomes and found a noncausal major duplication event, but we failed to explain this as clearly as we should have done. At the reviewers' suggestion, we looked more carefully at read depth, particularly for amplification of the hexose transporters and *SUC2*. We found one case of amplification at these loci: the *HXT6/HXT7* locus was duplicated in EvoClone7C. This amplification was also present in the backcrossed strain, indicating that it may have been causal. We now have the following paragraph in the Results section describing the three total events we found:

“We analyzed read depth across all evolved clones and saw three major duplication or deletion events: (1) Chromosome 3 in EvoClone10 was duplicated in the region between 152 and 172 kb from the left end of the chromosome. This doubling of read depth was reduced to a 50% increase in the segregated pool, indicating that the duplication was not causal. (2) The number of P-Type ATPases at the *ENO1/2/5* locus was reduced from 3 to 2 in EvoClone6 and EvoClone10. This reduction was also present in the backcrossed strain, indicating that it may have been causal. (3) The number of hexose transporter gene repeats at the *HXT6/7* locus increased from two in the ancestral strain to three in EvoClone7C. This amplification was also present in the backcrossed strain, indicating that it also may have been causal. Amplification of the *HXT6/7* locus has been found in other experimental evolutions (Brown et al. 1998; Gresham et al. 2008; Kao and Sherlock 2008).”

*5) Many wild* S. cerevisiae *strains are clumpy and loss of clumpiness is likely a lab-adapted trait. The authors should make it clear that loss and gain of clumpiness is likely to have occurred many times in yeast populations and provide other explanations for why clumpiness in yeast cells might be recurrently selected*.

We agree and we have added the following text to the discussion:

“Although lab strains of yeast were selected to be unicellular during their domestication (Mortimer and Johnston 1986), the level of clumpiness varies in wild isolates of yeast, and undifferentiated multicellularity may have been recurrently selected for its advantage in harvesting nutrients or for protection from stress (Smukalla et al. 2008).”